



# Global Storm Tide Modeling with ADCIRC v55: Unstructured Mesh Design and Performance

William J. Pringle[1], Damrongsak Wirasaet[1], Keith J. Roberts[2], and Joannes J. Westerink[1]

[1]Department of Civil and Environmental Engineering and Earth Sciences, University of Notre Dame, IN, USA
[2]School of Marine and Atmospheric Science, Stony Brook University, NY, USA.

**Correspondence:** William J. Pringle (wpringle@nd.edu)

**Abstract.** This paper details and tests numerical improvements to ADCIRC, a widely used finite element method shallow water equation solver, to more accurately and efficiently model global storm tides with seamless local mesh refinement in storm landfall locations. The sensitivity to global unstructured mesh design was investigated using automatically generated triangular meshes with a global minimum element size (MinEle) that ranged from 1.5 km to 6 km. We demonstrate that refining resolution based on topographic seabed gradients and employing a MinEle less than 3 km is important for the global accuracy of the simulated astronomical tide. Our recommended global mesh design (MinEle = 1.5 km) based on these results was locally refined down to two separate MinEle (500 m and 150 m) at the coastal landfall locations of two intense storms (Hurricane Katrina and Super Typhon Haiyan) to demonstrate the model's capability for coastal storm tide simulations and to test the sensitivity to local mesh refinement. Simulated maximum storm tide elevations closely follow the lower envelope of observed high water marks (HWMs) measured near the coast. In general, peak storm tide elevations along the open coast are decreased and the timing of the peak occurs later with local coastal mesh refinement. However, this mesh refinement only has a significant positive impact on HWM errors in straits and inlets narrower than the MinEle, and in bays and lakes separated from the ocean by these passages. Lastly, we demonstrate that the computational performance of the new numerical treatment is one-to-two orders of magnitude faster than studies using previous ADCIRC versions because gravity-wave based stability constraints are removed allowing for larger computational time steps.

## 1 Introduction

Extreme coastal sea levels and flooding driven by storms and tsunamis can be accurately modeled by the shallow water equations (SWEs). The SWEs are often numerically solved by discretizing the continuous equations using unstructured meshes with either finite volume methods (FVM) or finite element methods (FEM). These unstructured meshes can efficiently model the large range in lengthscales associated with physical processes that occur in the deep ocean to the nearshore region (e.g. Chen et al., 2003; Westerink et al., 2008; Zhang et al., 2016; Le Bars et al., 2016; Fringer et al., 2019), although many difficulties for large-scale ocean general circulation modeling remain (Danilov, 2013). However, for barotropic flows that are largely responsible for extreme coastal sea levels, the capability to model the global scale concurrently with local coastal scales resolved in sufficient detail so that emergency planning and engineering decisions can be made is well within reach. Further-





more, barotropic global storm tide models can be used as components of Earth System Models to analyze risks posed by the long-term response of extreme sea level and coastal flooding to climate change in far greater detail than currently possible (Bouwer, 2018; Vousdoukas et al., 2018).

A key practical advantage of ocean models discretized using FEM as compared to FVM is that they are usually less sensitive to mesh quality (e.g., element skewness). Specifically, ocean models discretized using FMV often use staggered C-grid

arrangements (e.g., Delft-FM) that have strict grid orthogonality requirements for numerical accuracy (Danilov, 2013; Fringer et al., 2019). The orthogonal requirement makes mesh generation over wide areas with fractal shoreline boundaries an arduous task and is difficult to automate, although progress has been made (Herzfeld et al., 2020; Hoch et al., 2020). Despite the difficulties, the FVM Delft-FM (Flexible Mesh) based Global Tide and Surge model (GTSM) (Verlaan et al., 2015), has been meticulously developed and widely used to generate reanalysis datasets, describe historical trends, and make projections of

extreme sea levels (Muis et al., 2016; Vousdoukas et al., 2018; Muis et al., 2019; Dullaart et al., 2019). The minimum (coastal) resolution of GTSM has been historically limited to ∼5 km but recently upgraded to 2.5 km (1.25 km in Europe) (Dullaart et al., 2019).

In the absence of constraints on orthogonality or element skewness, automatically generating unstructured triangular meshes on the spherical Earth that accurately conform to the coastline and cover a wide-range of spatial sales [O(10 m)-O(10 km)] is

completely realisable (Legrand et al., 2000; Gorman et al., 2006; Lambrechts et al., 2008; Roberts et al., 2019a). In one study, automatically generated ocean basin-scale meshes with variable element sizes (50 m to 10 km) were used to conduct dozens of numerically stable FEM simulation experiments without mesh hand-edits or numerical limiters (Roberts et al., 2019b). The ability of FEM models to handle rapid transitions in mesh element sizes combined with the ease of mesh generation with modern technologies (e.g., Roberts et al., 2019a) enables the application of seamless local refinement directly into the global

mesh where required, potentially on-the-fly based on atmospheric and ocean conditions that indicate a risk of coastal flooding.

This study conducts a systematic analysis of unstructured mesh design in order to assess and demonstrate the capabilities of global storm tide modeling using FEMs across multi-resolution scales spanning from the deep ocean to the nearshore coastal ocean environment. One outcome of this study is a recommendation of unstructured triangular-element mesh design of the global ocean that represents the barotropic physics with high fidelity using relatively few elements (Sect. 3.1). Moreover, we

test the added benefit of seamless local refinement in storm landfall regions to the simulation of storm tides (Sect. 3.2). Mesh generation is handled by the OceanMesh2D toolbox (Roberts et al., 2019a) which provides the tools to explore the effects of mesh design in a systematic way. For simulation we use the ADvanced CIRCulation (ADCIRC) model (Luettich and Westerink, 2004), which has been updated in this study for efficiency and to correctly model the SWEs on the sphere (Sect. 2.1); set for release as version 55. Section 3.3 summarizes the timing results with ADCIRC v55, highlighting its computational efficiency.





## 2 Methods and Experiment

### 2.1 Global Finite Element Storm Tide Model

The ADCIRC storm tide model used in this study is a FEM solver that has been extensively used for detailed hurricane inundation studies at local or regional scales (e.g., Westerink et al., 2008; Bunya et al., 2010; Hope et al., 2013), and as an operational storm tide forecast model run by the U.S. National Oceanic and Atmospheric Administration (NOAA) (Funakoshi et al., 2011; Vinogradov et al., 2017). ADCIRC solves the SWEs that are composed of primitive continuity and non-conservative depth-averaged momentum equations under astronomical and atmospherical forcing. After neglecting radial velocity terms, we formulate these equations in spherical coordinates as follows (Kolar et al., 1994),

$$\frac{\partial \zeta}{\partial t} = -\frac{1}{R\cos\phi}\left[\frac{\partial(UH)}{\partial\lambda} + \frac{\partial(VH\cos\phi)}{\partial\phi}\right] \tag{1}$$

$$\frac{\partial U}{\partial t} + \frac{g}{R\cos\phi}\frac{\partial\zeta}{\partial\lambda} = -\frac{U}{R\cos\phi}\frac{\partial U}{\partial\lambda} - \frac{V}{R}\frac{\partial U}{\partial\phi} - \frac{1}{R\cos\phi}\frac{\partial\Psi}{\partial\lambda} - (\mathcal{C}_{\lambda\phi} - f')V + \frac{\tau_w U_w}{\rho_0 H} - \left(\frac{\tau_b}{\rho_0 H} + \mathcal{C}_{\lambda\lambda}\right)U$$

$$+ \frac{1}{RH}\left[\frac{1}{\cos\phi}\frac{\partial\tau_{\lambda\lambda}}{\partial\lambda} + \frac{\partial\tau_{\lambda\phi}}{\partial\phi} - \tan\phi(\tau_{\lambda\phi} + \tau_{\phi\lambda})\right] \tag{2}$$

$$\frac{\partial V}{\partial t} + \frac{g}{R}\frac{\partial\zeta}{\partial\phi} = -\frac{U}{R\cos\phi}\frac{\partial V}{\partial\lambda} - \frac{V}{R}\frac{\partial V}{\partial\phi} - \frac{1}{R}\frac{\partial\Psi}{\partial\phi} - (\mathcal{C}_{\phi\lambda} + f')U + \frac{\tau_w V_w}{\rho_0 H} - \left(\frac{\tau_b}{\rho_0 H} + \mathcal{C}_{\phi\phi}\right)V$$

$$+ \frac{1}{RH}\left[\frac{\partial\tau_{\phi\phi}}{\partial\phi} + \frac{1}{\cos\phi}\frac{\partial\tau_{\phi\lambda}}{\partial\lambda} + \tan\phi(\tau_{\lambda\lambda} - \tau_{\phi\phi})\right] \tag{3}$$

where $\zeta$ is the free surface, $U$ and $V$ are the depth-averaged zonal and meridional velocities respectively, $H$ is the total water depth, $t$ is time, $\lambda$ is longitude, $\phi$ is latitude, $R$ is the radius of the Earth, and $\rho_0$ is the reference density of water. Additional terms are defined as follows,

$$\Psi = \frac{p_s}{\rho_0} - g\eta \quad : \text{external pressure and astronomical forcing}$$

$$f' = 2\Omega\sin\phi + \frac{\tan\phi}{R}U \quad : \text{Coriolis} + \text{component of advection expanded in spherical coordinates}$$

$$\tau_w = \rho_s C_D\sqrt{U_w^2 + V_w^2} \quad : \text{quadratic surface stress due to winds}$$

$$\tau_b = \rho_0 C_f\sqrt{U^2 + V^2} \quad : \text{quadratic bottom stress due to friction}$$

$$\mathcal{C} = \begin{pmatrix} \mathcal{C}_{\lambda\lambda} & \mathcal{C}_{\lambda\phi} \\ \mathcal{C}_{\phi\lambda} & \mathcal{C}_{\phi\phi} \end{pmatrix} \quad : \text{internal wave drag tensor}$$

$$= C_{it}\frac{\left[(N_b^2 - \omega^2)(N_m^2 - \omega^2)\right]^{1/2}}{4\pi\omega}\begin{pmatrix} (\nabla h_\lambda)^2 & \nabla h_\lambda\nabla h_\phi \\ \nabla h_\lambda\nabla h_\phi & (\nabla h_\phi)^2 \end{pmatrix} \quad \text{[local generation formulation]}$$





$$\tau = \begin{pmatrix} \tau_{\lambda\lambda} & \tau_{\lambda\phi} \\ \tau_{\phi\lambda} & \tau_{\phi\phi} \end{pmatrix} \quad : \text{lateral stress tensor}$$


$$= \begin{pmatrix} 2\nu_t H \frac{1}{R\cos\phi} \frac{\partial U}{\partial \lambda} & \nu_t H \frac{1}{R\cos\phi} \frac{\partial V}{\partial \lambda} \quad \text{or} \quad \nu_t H \left[ \frac{1}{R} \frac{\partial U}{\partial \phi} + \frac{1}{R\cos\phi} \frac{\partial V}{\partial \lambda} \right] \\ \nu_t H \frac{1}{R} \frac{\partial U}{\partial \phi} \quad \text{or} \quad \nu_t H \left[ \frac{1}{R} \frac{\partial U}{\partial \phi} + \frac{1}{R\cos\phi} \frac{\partial V}{\partial \lambda} \right] & 2\nu_t H \frac{1}{R} \frac{\partial V}{\partial \phi} \end{pmatrix}$$

where $p_s$ is the surface air pressure, $g$ is the gravitational acceleration, $\eta$ is the summation of the equilibrium tidal potential and self-attraction and loading (SAL) tide (Ray, 1998), $\Omega$ is the angular speed of the Earth, $R$ is the radius of the spherical Earth, $C_D$ is the quadratic wind drag coefficient, $\rho_s$ is the density of air at the ocean surface, $U_w$ and $V_w$ are the zonal and meridional 10-m wind velocities respectively, and $C_f$ is the quadratic bottom friction coefficient. $\mathcal{C}$ is the internal wave drag

tensor that accounts for the energy conversion from barotropic to baroclinic modes through internal tide generation in the deep ocean (Garrett and Kunze, 2007). Here, the local generation formulation is used (cf. Pringle et al., 2018a, b), in which $C_{it}$ is a global tuning coefficient, $N_b$ and $N_m$ are the seabed and depth-averaged buoyancy frequencies respectively, $\omega$ is set to the angular frequency of the M$_2$ tide, and $\nabla h_\lambda$, $\nabla h_\phi$ are the zonal and meridional topographic gradients respectively. Last, $\tau$ denotes the lateral stress tensor with $\nu_t$ denoting the lateral mixing coefficient. The components $\tau_{\lambda\phi}$ and $\tau_{\phi\lambda}$ can be chosen to

be either symmetric or non-symmetric as desired (Dresback et al., 2005). For this study we choose the symmetric option.

To properly compute the governing equations on the spherical Earth in the FEM framework used by ADCIRC, we have upgraded the model formulation and code as detailed in Sect. S1 of the supplementary document. This involves rotating the Earth so that the pole singularity is removed (Sect. S1.3) before applying a rectilinear mapping projection to transform the governing equations into a Cartesian form with spherical-based corrections to the spatial derivatives (Sect. S1.1). Here, the

continuity equation is multiplied by a factor dependent on the choice of cylindrical projection used (e.g., Mercator) to produce a conservative form that leads to discrete mass conservation and stability (cf. Hervouet, 2007; Castro et al., 2018) (Sect. S1.2). The stability of the ADCIRC solution scheme (Sect. S1.4) was analyzed in one-dimensional linear form (Sect. S1.5) to provide guidelines for the choice of numerical parameters that can be chosen to remove the gravity-wave based (CFL) constraint. The validity of this analysis in the 2D nonlinear form has been demonstrated through the numerical simulations presented in this

paper. With a semi-implicit time integration scheme, computational time steps (up to 120s) permitted are much larger than the CFL constraint and as a result facilitate computationally efficient global simulations. Hereafter, the updated code in this study refers to the new release, ADCIRC v55. Solutions using the uncorrected model formulation are referred to by the previous version, ADCIRC v54.

## 2.2 Unstructured Triangular Mesh Generation on the Earth

The global unstructured meshes in this study are generated automatically using scripts with Version 3.0.0 of OceanMesh2D (Roberts et al., 2019a; Pringle and Roberts, 2020). No post-processing hand-edits of any mesh were necessary to facilitate numerically stable simulations. Meshes are built in the stereographic projection centered at the North Pole to maintain angle conformity on the sphere and have the elements wrap around the Earth seamlessly including an element placed over the North





**Table 1.** The mesh size functions used to spatially distribute element resolution, $E_R$. The variable parameter in each function is indicated by $\alpha$.

| Code | Long Name | Function Expression |
|------|-----------|---------------------|
| MinEle | Nominal minimum element size bound | $E_R \geq \alpha$ |
| D | Nearest distance-to-shoreline | $E_R = \text{MinEle} + \alpha d_s$ |
| WL | Wavelength-to-element size ratio | $E_R = \frac{T_{M_2}}{\alpha}\sqrt{gh}$ |
| TLS | Topographic-length-scale (using filtered topographic gradients) | $E_R = \frac{2\pi}{\alpha}\frac{h}{|\nabla h^*|}$ |
| FL | Low-pass filter length (fraction of barotropic Rossby radius) | $h^* = \mathcal{F}_{lp}(L) * h, \quad L = \alpha\frac{\sqrt{gh}}{f}$ |
| G | Nominal element-to-element gradation limit on resolution | $\Rightarrow |\nabla E_R| < \alpha$ |
| MaxEle | Nominal maximum element size bound | $E_R \leq \alpha$ |

$d_s$: shortest distance to the shoreline, $T_{M_2}$: period of the $M_2$ tidal wave, $h$: still-water depth,

$h^*$: low-pass filtered $h$, in which $\mathcal{F}_{lp}(L)$ is the low-pass filter with cutoff length, $L$.

Pole (Lambrechts et al., 2008). Interested readers can execute "Example_7_Global.m" contained within the OceanMesh2D
package to generate their own global mesh in a similar fashion.

Mesh design is handled through mesh size (resolution distribution) functions that are defined on a regular structured grid, usually that of the topo-bathymetric digital elevation model (DEM). In this study we use functions based on distance-to-shoreline, bathymetric depths, and topographic gradients (see Table 1 for definitions). The final mesh size function is found by taking the minimum of all individual functions, and applying nominal minimum and maximum mesh resolution bounds and a
element-to-element gradation limiter to bound the transition rate (Roberts et al., 2019a). The effects of the individual mesh size functions and bounds on barotropic tides in a regional model have been previously detailed in Roberts et al. (2019b), which we use to guide our experiments exploring mesh design.

Additionally, this study makes use of the OceanMesh2D "plus" function which seamlessly merges two arbitrary meshes together keeping the finer resolution in the overlapping region. We use this function to apply local mesh refinement to a global
mesh in storm-affected coastal regions to better resolve semi-enclosed bays and lakes, inlets, backbays, channels, and other small-scale shoreline geometries.

## 2.3 Experimental Design

The experimental design we pursue is composed of two distinct steps, both with the purpose to maximize model efficiency while maintaining a threshold of accuracy. First, we begin with a mesh design that we assume is a highly-refined discretization
of the Earth in a global sense, and systematically relax the mesh size parameters to reduce the total number of mesh vertices while trying to minimize any negative impacts on global model accuracy. Second, we take the resulting recommended global mesh design from the previous step, and apply local mesh refinement to increase the coastal resolution in the storm landfalling region and potentially improve local model accuracy.





**Table 2.** Summary of the global mesh designs. Each row is separate and made up of three mesh designs for each variable mesh size function parameter, in addition to the Ref mesh design which is the same for each row.

| Variable Mesh Size Function Parameter | Design Code | | | | Other Mesh Size Function Parameter Values |
|---|---|---|---|---|---|
| | Ref | A | B | C | |
| MinEle | 1.5 km | 2.25 km | 3 km | 6 km | MaxEle = 25 km, WL = 30, D = G = 0.35, TLS = 20, FL = 0 |
| TLS | 20 | 10 | 5 | 0 [not used] | MinEle = 1.5 km, MaxEle = 25 km, WL = 30, D = G = 0.35, FL = 0 |
| FL | 0 [not used] | 1/80 | 1/20 | 1/5 | MinEle = 1.5 km, MaxEle = 25 km, WL = 30, D = G = 0.35, TLS = 20 |

### 2.3.1 Step 1: Global Mesh Design

In this step, three mesh size function parameters (MinEle, TLS, and FL) are systematically relaxed to coarsen an initially highly-refined discretization of the Earth, termed the reference (Ref) mesh design (c.f., Table 2). In the Ref design, MinEle is set to 1.5 km mainly due to practical constraints (e.g., computer memory usage when generating a global mesh in OceanMesh2D), and MaxEle is set to 25 km because common global meteorological products are defined on ∼25 km grids (aliasing errors in interpolation of the meteorological input data becomes an issue when a coarser grid is considered). The parameters WL, D,

and G are considered fixed for all mesh designs based on prior knowledge that ∼30 elements per wavelength (WL) is sufficient for global ocean tides (Greenberg et al., 2007), and an element-to-element gradation limit/distance expansion rate (G and D) as high as 0.35 is tolerable as long as the TLS mesh size function is applied (Roberts et al., 2019b). As a note, the FL parameter is used in the construction of the TLS mesh size function to filter out small-scale topographic features (c.f., Table 1) that are potentially unimportant to the local barotropic physics, thus disregarding those features for the application of higher resolution.

This study is the first to test the effect of the FL parameter in detail. In each mesh design perturbation, only one of the three parameters is changed while the other parameters are kept identical to those used in the Ref design. For illustration, the spatial distribution of resolution of the Ref design compared to the TLS-B design is shown in Fig. 1.

To assess the effect on the global model accuracy as the mesh designs are coarsened, we compare simulated astronomical tidal solutions to the data-assimilated TPXO9-Atlas. We focus on astronomical tides in this step because they can be reduced

to a series of harmonic constituents of well-defined frequencies to make systematic global comparisons (Roberts et al., 2019b; Pringle et al., 2018a). The TPXO9-Atlas is the latest release of the TPXO satellite-assimilated tidal model (Egbert and Erofeeva, 2002). According to Egbert and Erofeeva (2019), the mean $M_2$ $RMSE_t$ (tidal root-mean-square error, see Appendix A for definition) is 0.5 cm versus Stammer et al. (2014) deep ocean tide gauges and ∼3 cm versus Stammer et al. (2014) shallow water tide gauges. The metric of comparison between mesh designs is based on the area-weighted empirical cumulative distri-

bution function (ECDF) of the 5-constituent total tidal root-mean-square error, $RMSE_{t|tot}$ (c.f., Appendix A). The two-sample Kolmogorov-Smirnov test statistic, $K$ (c.f., Appendix A) is used to provide a single metric of comparison between two ECDF curves.



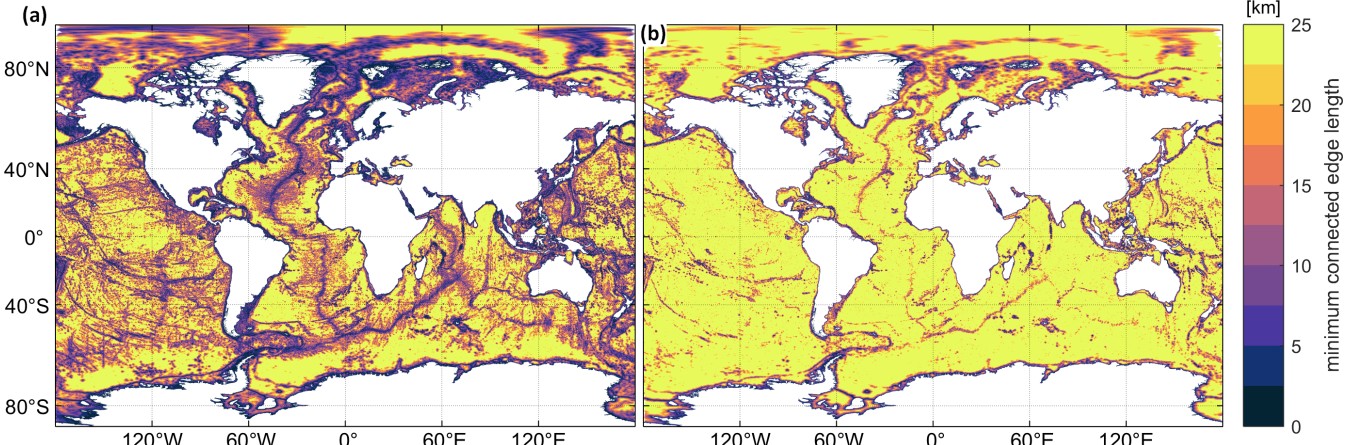

**Figure 1.** Mesh resolution distribution (defined as the minimum connected element edge length for a mesh vertex) for two global mesh designs. (a) Ref, (b) TLS-B (c.f., Table 2).

### 2.3.2 Step 2: Local Mesh Refinement

In this step, the recommended global mesh design from Step 1 (Sect. 2.3.1) is used but with additional patches of high resolution (local mesh refinement) near the landfall location of two storm events. We choose to focus on a particularly significant historical storm event from each of the Atlantic and Pacific ocean basins where tropical cyclones most commonly occur:

1. Western North Atlantic – Hurricane Katrina, August 23-31, 2005. The most severe impact of storm tide induced coastal flooding occurred in the Louisiana/Mississippi region of USA in the northern Gulf of Mexico (URS Group Inc, 2006a, b).

2. Western Pacific – Super Typhoon Haiyan, November 3-11, 2013. The most severe impact of storm tide induced coastal flooding occurred in and around Tacloban, Philippines at the back end of the Leyte Gulf (Mori et al., 2014).

Local mesh refinement is achieved by using OceanMesh2D to automatically merge a locally generated mesh for each landfall region (Louisiana/Mississippi: Fig. 2, Leyte Gulf: Fig. 3) into the global mesh. Two local meshes are generated for each region: one with MinEle = 500 m and another with MinEle = 150 m. The other mesh size function parameters are kept the same as the global mesh design, except that G and D are reduced to 0.25 because the mesh quality generated in the local domain was considered too low with a value of 0.35. Furthermore, the locally refined meshes only add an additional 0.5-3% to the total mesh vertex count compared to the original global mesh design, thus there is limited motivation to use higher values of G and D to try and save on mesh vertices in the local refinement region.

To assess the accuracy and inter-compare mesh designs, we primarily use the output of the maximum simulated storm tide elevation from each event. For validation we compare to surveys of high-water marks (HWMs) in the landfall regions that are located close to the coast (within 1.5 km of the nearest 150-m locally refined mesh vertex) and attributed primarily to surge

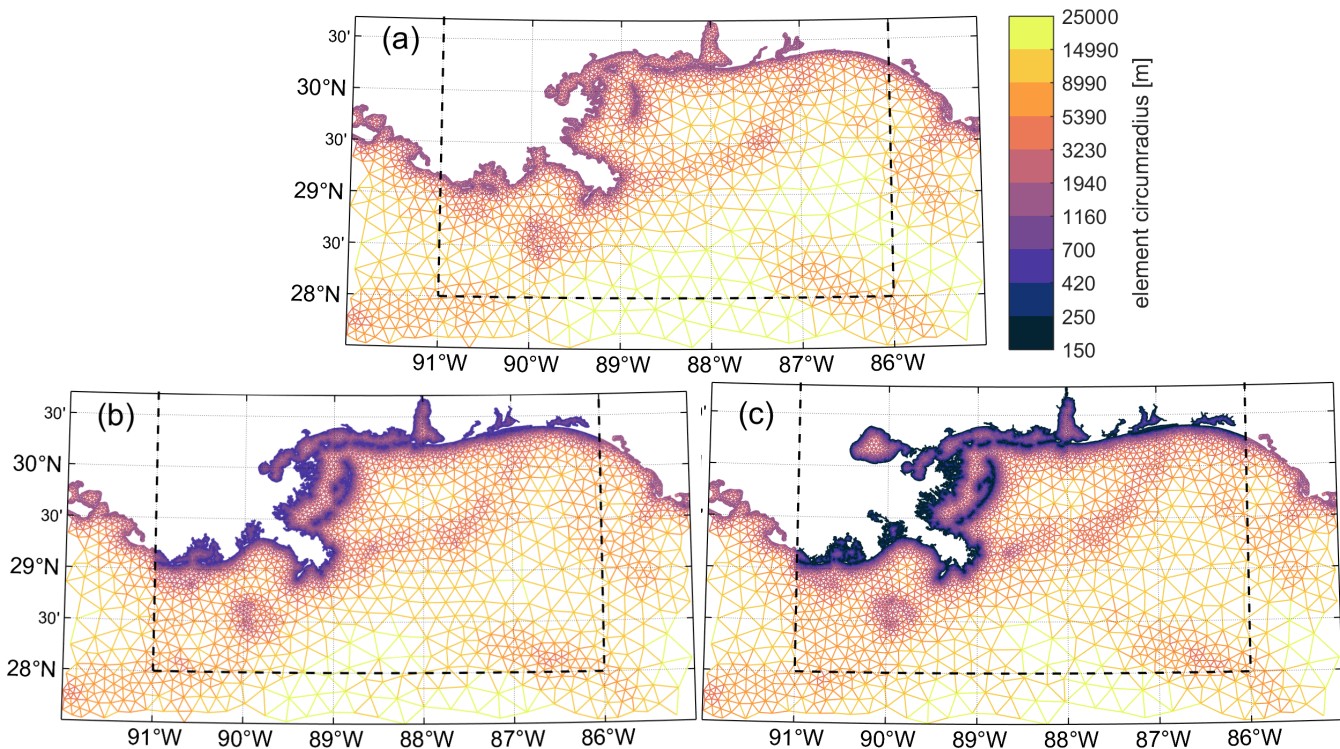

**Figure 2.** Comparisons of mesh triangulation and resolution (defined as the element circumradius on a Lambert conformal conic projection) in the Hurricane Katrina landfall region around Louisiana/Mississippi, USA. (a) MinEle = 1.5 km (default TLS-B global mesh design), (b) MinEle = 500 m local mesh refinement, (c) MinEle = 150 m local mesh refinement. The black dashed boxes indicate where the local mesh refinement was applied.

for both Katrina (URS Group Inc, 2006a, b) and Haiyan (Mori et al., 2014). Note that for Katrina we add a value of 0.23 m to the simulated storm tide elevations to account for a steric offset and the conversion to NAVD88 vertical datum from local mean sea level (Bunya et al., 2010). No adjustment is made for Haiyan. The standard RMSE, the mean absolute error

(MAE) and its standard deviation (SD) are reported. In addition, for Katrina we plot the storm tide time series signal at three coastal NOAA tide gauges with available historical data (IDs: 8735180, 8743281, 8761724). For Haiyan, no reliable time series observations of the main event are available (Mori et al., 2014), so we compare to the astronomical tide reconstructed from TPXO9-Atlas constituents at three selected locations for reference. The geographical location of the HWMs, tide gauges, and selected locations are shown with the results in Sect. 3.2.

**2.4   Datasets and Model Setup**

The full-resolution Global Self-consistent Hierarchical High-resolution Shorelines (GSHHS) dataset (Wessel and Smith, 1996) is used to define the shoreline boundary when making the mesh. The most recent global bathymetry DEM, GEBCO_2019

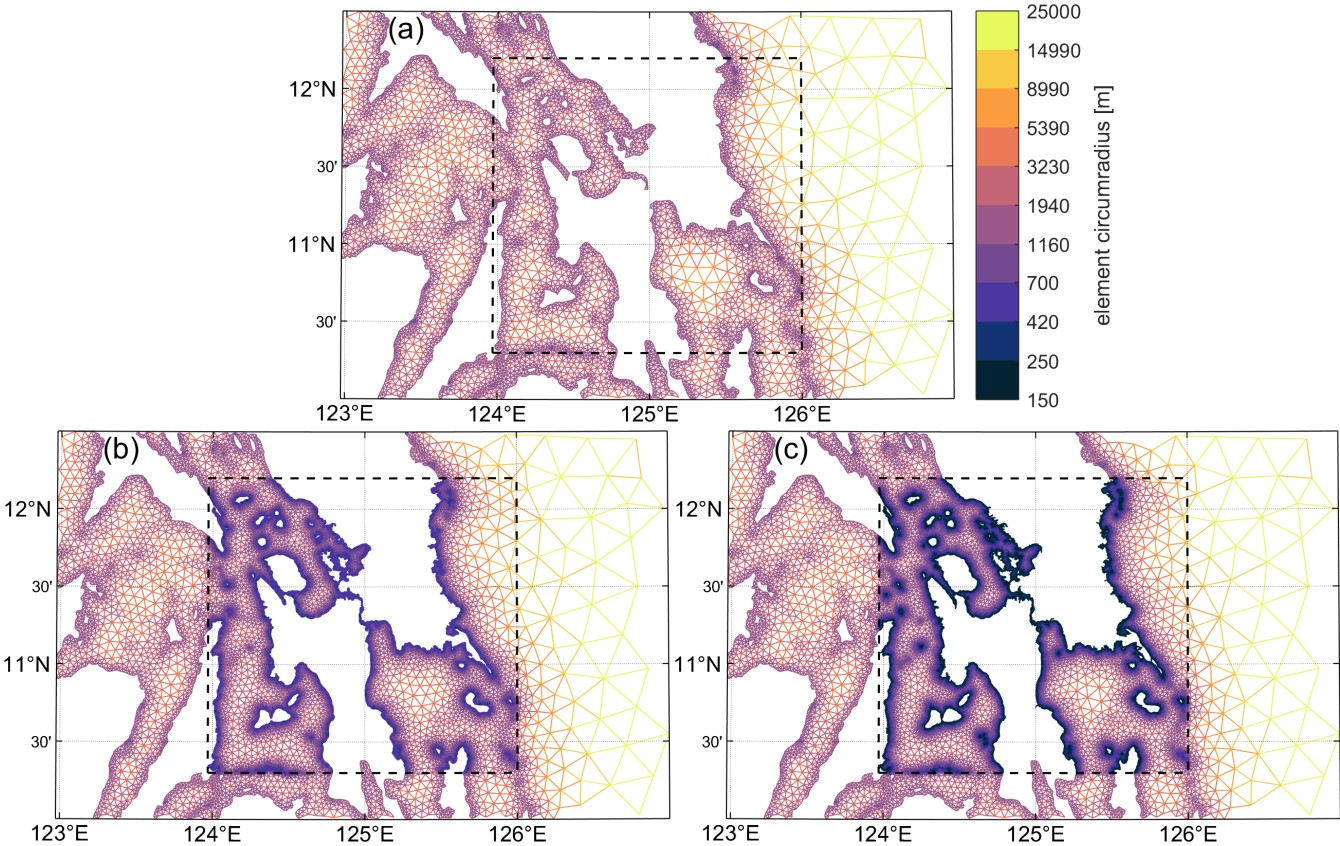

**Figure 3.** Comparisons of mesh triangulation and resolution (defined as the element circumradius on a Lambert conformal conic projection) in the Super Typhoon Haiyan landfall region around Leyte and Samar Island, Philippines. (a) MinEle = 1.5 km (default TLS-B global mesh design), (b) MinEle = 500 m local mesh refinement, (c) MinEle = 150 m local mesh refinement. The black dashed boxes indicate where the local mesh refinement was applied.

(GEBCO Compilation Group, 2019), which has an equatorial resolution of ∼500 m, is used to prescribe the bathymetry for the model (Sect. S2.2). Underneath Antarctica ice shelves, the RTopo-2 DEM (Schaffer et al., 2016), which has an equatorial

resolution of ∼1 km, is used to prescribe ocean depths taking into account the ice shelf thickness. The SAL tide is specified from the FES2014 (Lyard et al., 2006) data assimilated tidal solutions (Sect. S2.4). The buoyancy frequency data required to compute the internal wave drag tensor, $\mathcal{C}$ (Sect. S2.6) is calculated from the 2005-2017 decadal average of salinity and temperature data taken from the World Ocean Atlas 2018 (Locarnini et al., 2019; Zweng et al., 2019).

Atmospheric forcings are derived from three different sources in this study (see Sect. S2.5 for details on how to use each

source in ADCIRC v55). Hourly global reanalysis datasets, 0.313° CFSR (Saha et al., 2010) for Katrina and 0.205° CFSv2 (Saha et al., 2014) for Haiyan, are used outside of the local storm regions (Fig. 4a). Locally in the Gulf of Mexico region, Hurricane Katrina is forced by 15-min 0.050° OceanWeather Inc. (OWI) atmospheric reanalysis data (Bunya et al., 2010) from



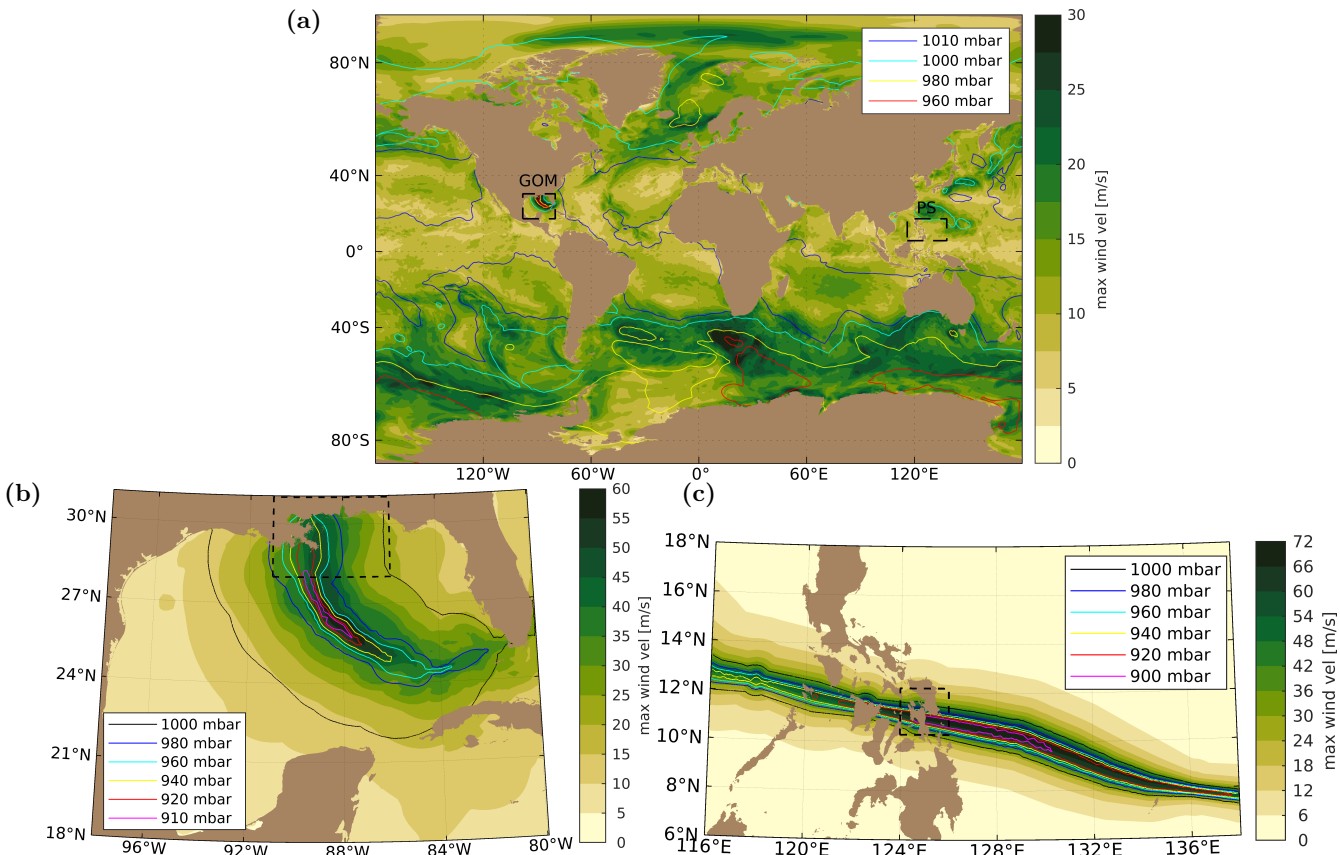

**Figure 4.** Maximum 10-m wind velocities and minimum pressure contours over the ocean for Hurricane Katrina (August 25-31, 2005) and Super Typhoon Haiyan (November 4-10, 2013). (a) Global view displaying CFSR reanalysis data during Katrina except in the Gulf of Mexico region [indicated by the black dashed box labeled GOM]; (b) Gulf of Mexico region displaying OWI reanalysis data of Katrina; (c) Philippine Sea region [indicated by the black dashed box labeled PS in (a)] displaying the best-track Holland parametric vortex model of Haiyan. Black dashed boxes in (b) and (c) indicate the landfall regions where the mesh is locally refined for each storm.

August 25 to August 31 (Fig. 4b). Super Typhoon Haiyan meteorology is described by the best-track Holland parametric vortex model (Holland, 1980) from November 4 to November 10 (Fig. 4c).

When only astronomical tides are simulated, we force the model with only astronomical forcing ($\eta$) for the five leading astronomical tidal constituents ($M_2$, $S_2$, $N_2$, $K_1$, $O_1$), and analyze for the corresponding constituent amplitude and phases using a 28-day harmonic analysis. These five constituents are chosen so that we can use a relatively short 28-day harmonic analysis period (Ngodock et al., 2016), which would otherwise need to be extended to around 180 days if other constituents are included because of the closeness in their frequencies (e.g., $K_1$ and $P_1$) (Pringle et al., 2018a). When simulating storm tides,

both atmospheric ($\tau_w$ and $p_s$) and astronomical forcings are invoked, this time using the following ten tidal potential-generating constituents to obtain a more complete tidal signal: $M_2$, $S_2$, $N_2$, $K_2$, $O_1$, $P_1$, $K_1$, $Q_1$, MF, MM. In both cases the model is spun-





**Table 3.** $\overline{\text{RMSE}}_t$ [cm] (c.f., Appendix A) values for simulated tidal results using ADCIRC v55 (upgrade) and ADCIRC v54 in deep ($h > 1$ km) and shallow ($h < 1$ km) waters on the Ref mesh. Results from other forward barotropic tidal models (Stammer et al., 2014; Ngodock et al., 2016; Schindelegger et al., 2018) are included for comparison where known.

|  | $M_2$ $\overline{\text{RMSE}}_t$ [cm] |  | $\overline{\text{RMSE}}_{t|tot}$ [cm] |  |
| --- | --- | --- | --- | --- |
| Model | Deep water | Shallow water | Deep water | Shallow water |
| ADCIRC v54 | 6.5 | 18.5 | 7.92 | 22.1 |
| ADCIRC v55 | 2.87 | 13.9 | 3.89 | 17.2 |
| Stammer et al. (2014)* | 5.25-7.76 | 18.6-27.9 | - | - |
| Ngodock et al. (2016)*# | 2.6-3.2 | - | - | - |
| Schindelegger et al. (2018)* | 4.4 | 14.6 | - | - |

\*: $\overline{\text{RMSE}}_t$ is computed against TPXO8-Atlas rather than TPXO9-Atlas.

\#: Uses state ensemble Kalman Filter (perturbed data assimilation).

up from a quiescent state for approximately four weeks to make sure that global tides are in relative equilibrium. Complete specification details to setup the ADCIRC v55 model simulations in this study are detailed in Sect. S2 of the supplementary document.

## 3 Results

### 3.1 Global Mesh Design

#### 3.1.1 Validation of the Reference Mesh

When simulating on the Ref mesh using ADCIRC v55, significant improvements to the prediction of astronomical tidal constituents were measured compared to ADCIRC v54. In particular, $M_2$ amphidromes in the high latitude regions are largely corrected such that any disparities between TPXO9-Atlas and our updated model solutions are qualitatively hard to discern from a global perspective (Fig. 5). Moreover, $M_2$ $\text{RMSE}_t$ is less than 2.5 cm over most of the ocean with the largest remaining deep ocean hotspot in the North Atlantic (Fig. 6). Indeed, deep ocean $M_2$ $\overline{\text{RMSE}}_t$ = 2.87 cm (Table 3), which is smaller than those previously computed for non-assimilated barotropic tidal models (Stammer et al., 2014; Schindelegger et al., 2018), and within the range of errors computed for solutions obtained by embedding a state ensemble Kalman Filter (perturbed data assimilation) into a forward ocean circulation model (Ngodock et al., 2016). Furthermore, the 5-constituent total tidal error, $\overline{\text{RMSE}}_{t|tot}$, is less than 4 cm in the deep ocean. In shallow regions, the $M_2$ $\overline{\text{RMSE}}_t$ is 13.9 cm, which is slightly smaller than obtained in Schindelegger et al. (2018), and $\overline{\text{RMSE}}_{t|tot}$ is 17.2 cm. Note that the area-weighted median value of $\text{RMSE}_{t|tot}$ (c.f., Appendix A) in shallow water is just 6.63 cm.



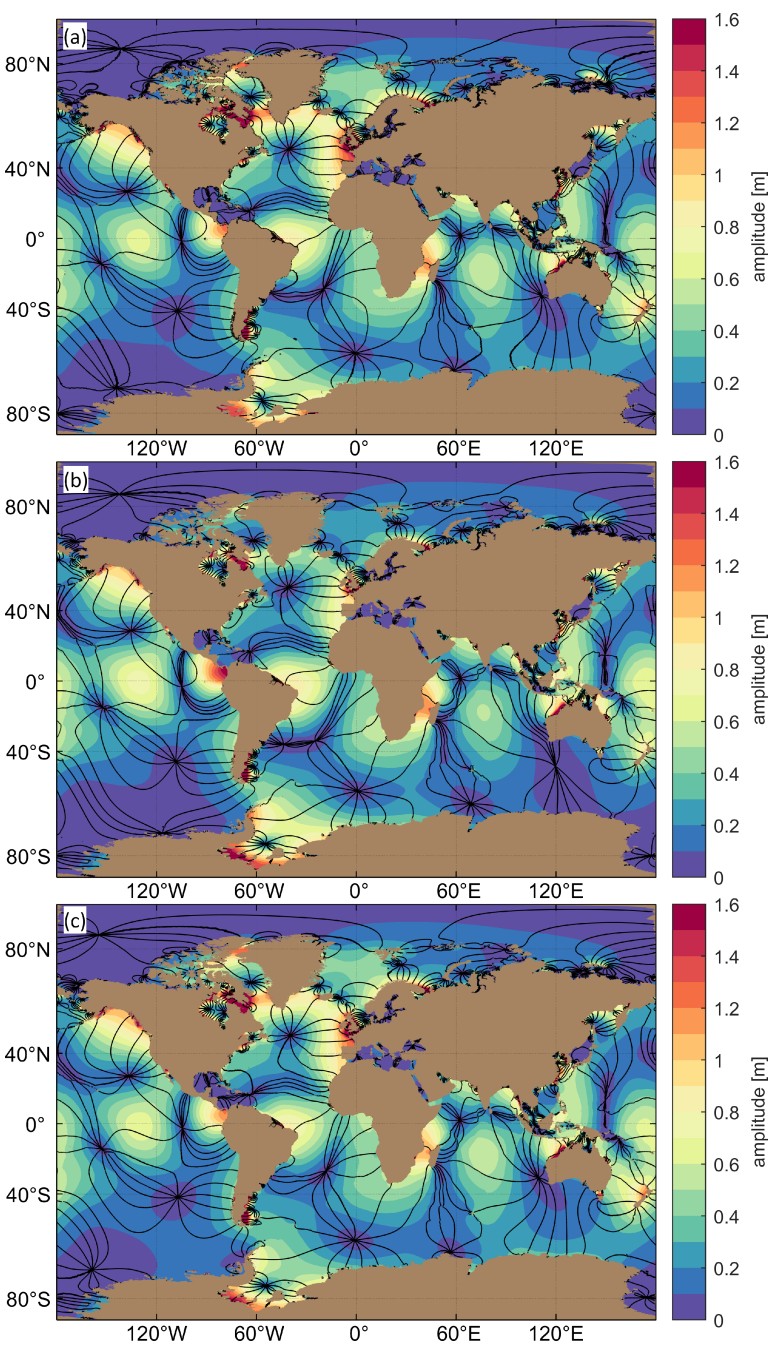

**Figure 5.** Global $M_2$ tidal amplitude and phase (cotidal lines are drawn in $30°$ increments) plots for (a) TPXO9-Atlas, (b) ADCIRC v54 on the Ref mesh, and (c) ADCIRC v55 upgrade on the Ref mesh.



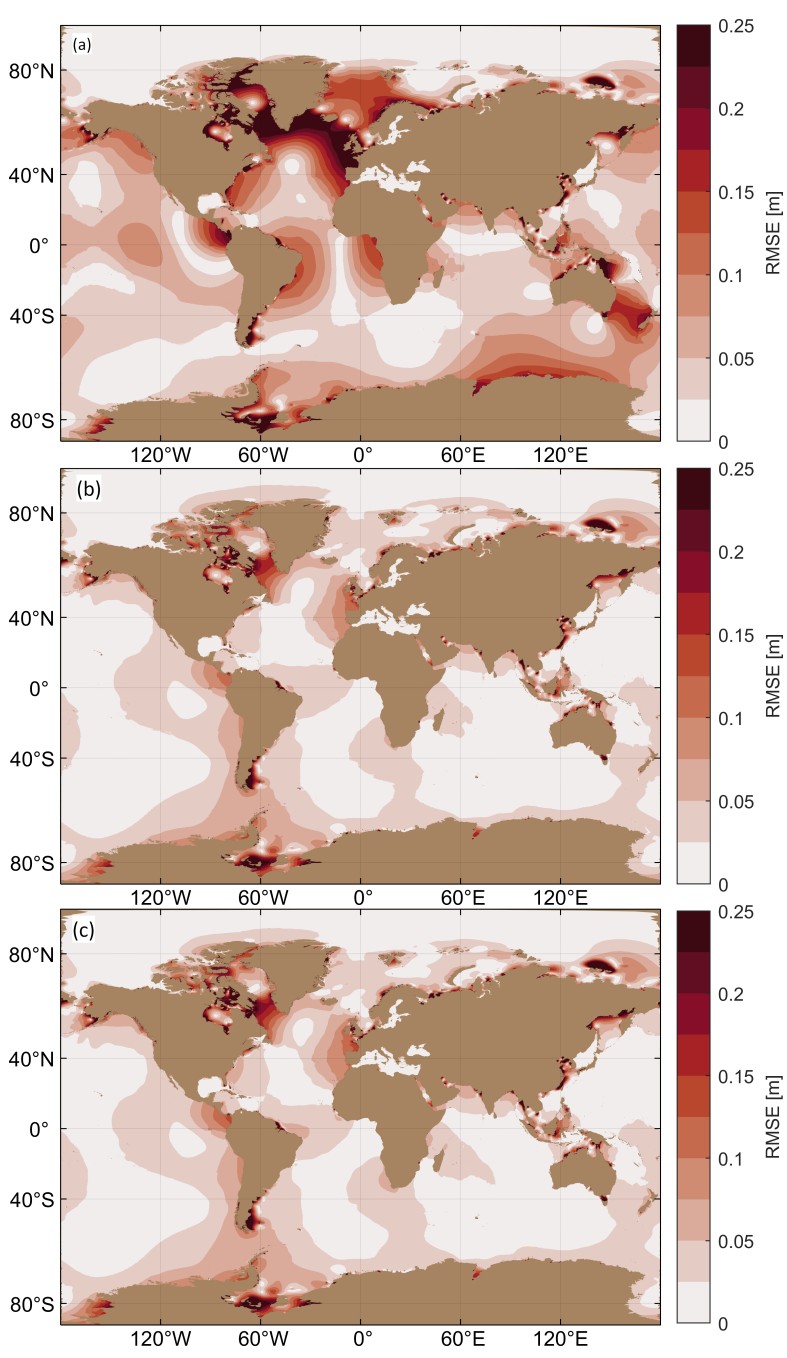

**Figure 6.** Global $M_2$ $RMSE_t$ computed against the TPXO9-Atlas for; (a) ADCIRC v54 on the Ref mesh, (b) ADCIRC v55 upgrade on the Ref mesh, and (c) ADCIRC v55 upgrade on the TLS-B mesh.

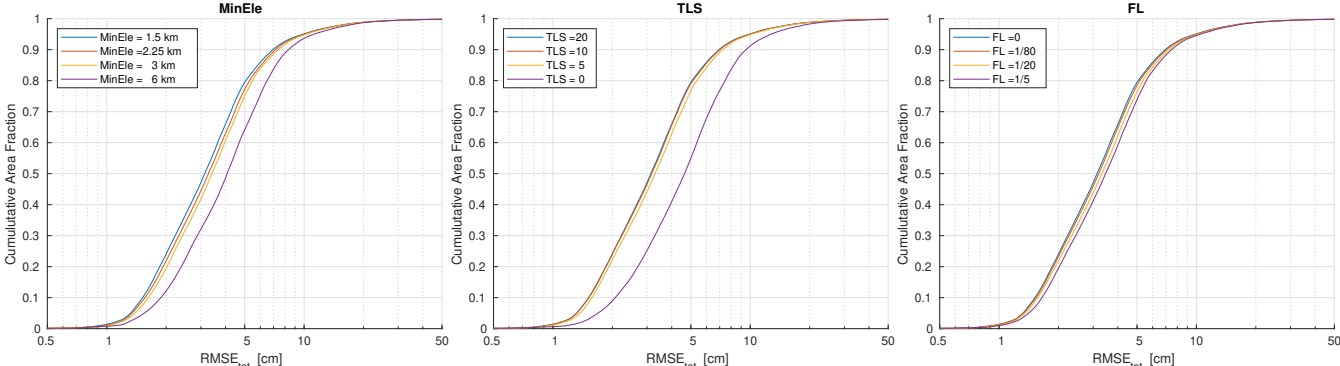

**Figure 7.** Area-weighted ECDF curves of RMSE$_{t|tot}$ for the global mesh designs varying MinEle, TLS, and FL (left-to-right). See Table 2 for mesh design details.

### 3.1.2 Solution Variability with Global Mesh Design Parameters

The distribution (ECDF curves) of RMSE$_{t|tot}$ degrades as the mesh size function parameters are relaxed (Fig. 7). Most of this degradation occurs in the body of the distribution rather than the tails. This characteristic implies that the $K$ test statistic is a good metric of disparity between mesh designs (Fig. 8) because it measures the greatest vertical distance between ECDF curves which has a greater chance of being larger in the body. $K$ is positive for all mesh designs indicating that the Ref mesh is indeed statistically the best performing mesh.

Increasing MinEle has a clear but gradual degenerative effect on the solution as it is increased from 1.5 km to 3 km. The disparities in the ECDF curves noticeably grow as MinEle is increased to 6 km; the value of $K$ increases from 0.057 for 3 km to 0.175 for 6 km. In comparison, the ECDF curves and the value of $K$ is changed comparatively little as the TLS parameter is decreased from 20 to 5 ($K$ is just 0.032 for TLS = 5). In fact, the solutions are close to identical for TLS values of 10 and 20. However, as the TLS function is turned off (TLS = 0), the solution is severely degraded and the value of $K$ is the greatest for any mesh design (= 0.257). The effect of using the FL function and relaxing the parameter on the ECDF curves is fairly gradual overall, however the magnitude of this change is not trivial; $K$ is increased to 0.070 for FL = 1/5.

In summary, the results demonstrate that decreasing the TLS parameter from 20 through to 5 substantially decreases the number of vertices while it has a relatively small effect on the tidal solution compared to the other experiments. On the other hand, increasing the FL parameter has a comparatively small impact on vertex count reduction, while increasing MinEle has a relatively large impact on the solution. The final choice of mesh design is dependent on one's tolerance for error, but in general it is preferable to choose a mesh that is plotted close to bottom-left corner of the graph in Fig. 8. Following this logic, the TLS-B mesh design (MinEle = 1.5 km, TLS = 5, FL = 0) appears to be the most efficient one tested (see Fig. 6 for the spatial distribution of M$_2$ RMSE$_t$ on this mesh). The results also suggest that combinations of MinEle larger than 1.5 km (∼2.25-3 km) and a TLS parameter smaller than 20 (∼5-10) could be employed to potentially lead to an efficient mesh design.



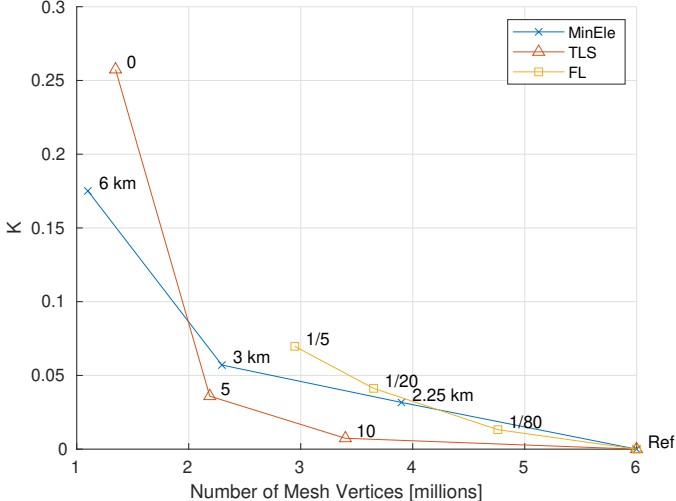

**Figure 8.** The two-sample Kolmogorov-Smirnov test statistic, $K$ versus the total number of mesh vertices. $K$ is computed as the largest vertical distance between the $\mathrm{RMSE}_{t|tot}$ ECDF curve of the Ref mesh design and the ECDF curve of the coarsened mesh designs (varying MinEle, TLS, and FL; Table 2).

Nevertheless, using a small MinEle is in and of itself useful to provide extra coastal resolution, which may be more important as we consider local storm tide accuracy. Thus, TLS-B is chosen as the base mesh design in Sect. 3.2.

### 3.2 Local Mesh Refinement

#### 3.2.1 Validation on the 150-m Locally-Refined Meshes

The maximum storm tide elevation due to Hurricane Katrina approaches 8 m in the Hancock and Harrison Counties of Mississippi (Fig. 9a), comparable to previously conducted high-fidelity simulation results (Dietrich et al., 2009). For Super Typhoon Haiyan, the maximum storm tide elevation exceeds 6 m near Tacloban due to local amplification in the Leyte Gulf (Fig. 9b), similar to previous simulation results (Mori et al., 2014). Qualitatively good agreement with the plotted HWMs is demonstrated for both storms with a few exceptions.

As pointed out by Mori et al. (2014), since inundation is not simulated and the effects of wave setup are ignored in these simulations we expect the maximum storm tide height to match the lower envelope of observed HWMs due to the amplification by topography. By numbering the HWMs starting at the most southwest point and following the shoreline clockwise, it is indeed illustrated that the simulation results tend to mach the lower envelope of observed HWMs quite closely (Fig. 10). For Katrina, the MAE = 1.06 m (SD = 0.66 m) (see Fig. 10 legend for error statistics) from 138 HWMs is larger than the MAE = 0.36 m (SD = 0.44 m) based on 193 HWMs in Bunya et al. (2010). However, their simulations included wave setup, river flows, levees, and dynamic inundation so our results can be considered quite reasonable in comparison. For Haiyan, the RMSE = 1.45 m from 145 HWMs closely matches the RMSE range of 1.29-1.45 m based on 60 HWMs in Mori et al. (2014).





**Figure 9.** Maximum simulated storm tide elevations on the MinEle = 150 m local refinement meshes compared to coastal surge attributed high water marks that are located within 1.5 km of the mesh coastline (filled circles). Black annotated crosses indicate the locations where storm tide time series are plotted. (a) Maximum simulated elevations due to Hurricane Katrina in the Louisiana/Mississippi landfall region. A value of 0.23 m has been added to account for a steric offset and the conversion to NAVD88 vertical datum from local mean sea level; (b) Maximum simulated elevations due to Super Typhoon Haiyan in the Leyte and Samar Island, Philippines landfall region.

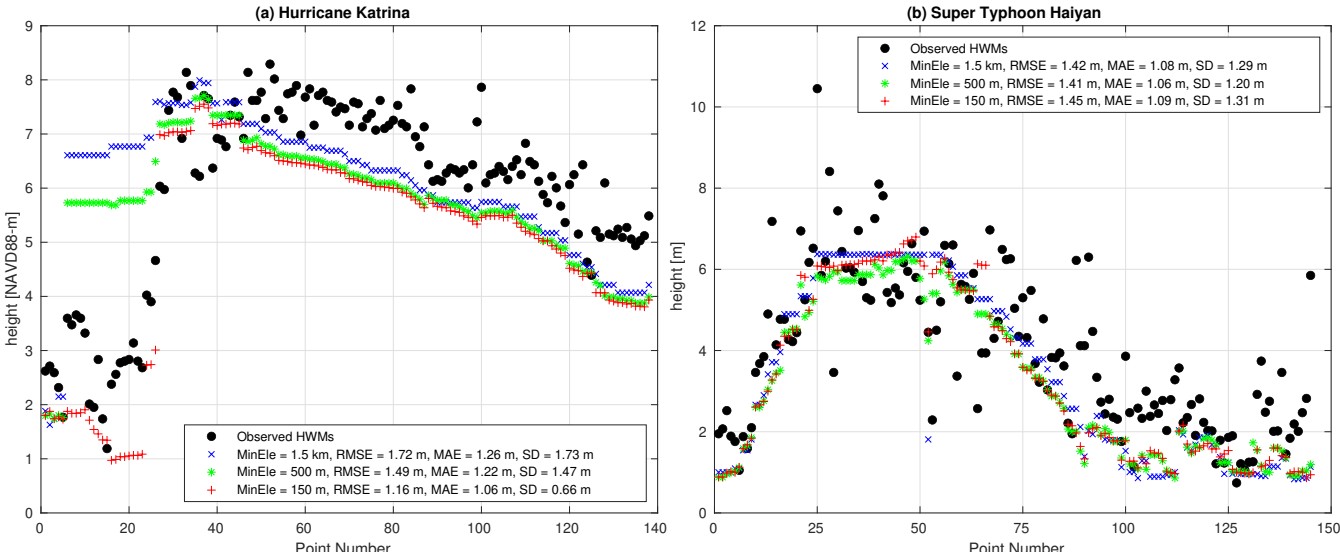

**Figure 10.** Comparisons of the observed HWMs to the simulated maximum storm tide height at the nearest vertex on meshes with different local MinEle (1.5 km, 500 m, 150 m). (a) Hurricane Katrina (0.23 m has been added to simulated results) and (b) Super Typhoon Haiyan. Points are numbered by starting from the most southwest point and following the shoreline clockwise around.

Time series of Hurricane Katrina at NOAA tide gauges show that the timing of the peak storm tide elevation and the amplitude and phase of the tide signal prior to landfall are well represented by the model (Fig. 11). The modeled peak is underestimated by ~0.4 m at gauge 8735180, and at gauge 8743281 where the largest peak storm tide occurred, tide gauge recording was interrupted as the storm was making landfall, but the simulation closely follows the observations up until this point. A roughly constant discrepancy of ~0.4-0.5 m between the simulation and observation develops at the gauges following the last high tide prior to storm landfall, likely explaining the underestimate at gauge 8735180 (by this logic the peak may be overestimated at 8761724). Comparing to time series of simulations with wind wave-coupling (Roberts and Cobell, 2017), it appears that this discrepancy can be attributed to wave setup effects that are ignored here. Time series of Super Typhoon Haiyan also show that the amplitude and phasing of the tide at the selected locations are fairly well represented as compared to the TPXO9-Atlas (Fig. 11). Storm tide heights at Tacloban are dominated by the short and intense surge event but the duration of surge is likely underestimated because the parametric vortex model lacks background winds. Due to the timing of the storm landfall during the lower high tide, peak storm tides exceeded the higher high tide by just 0.5 m at Guiuan. In fact for Guiuan and Canuay Island the minimum storm tide levels were more severe than the peak levels.

### 3.2.2 Solution Variability with Local Mesh Refinement

Our results indicate that there is a tendency for the coarser resolution meshes to have larger storm tide elevations in the open coastal areas of the landfall region. In the case of the Katrina, this is most clearly seen in Lake Borgne where maximum storm tide elevations are at least 0.6 m larger on the MinEle = 1.5 km mesh than those on the MinEle = 500 m mesh, and approximately



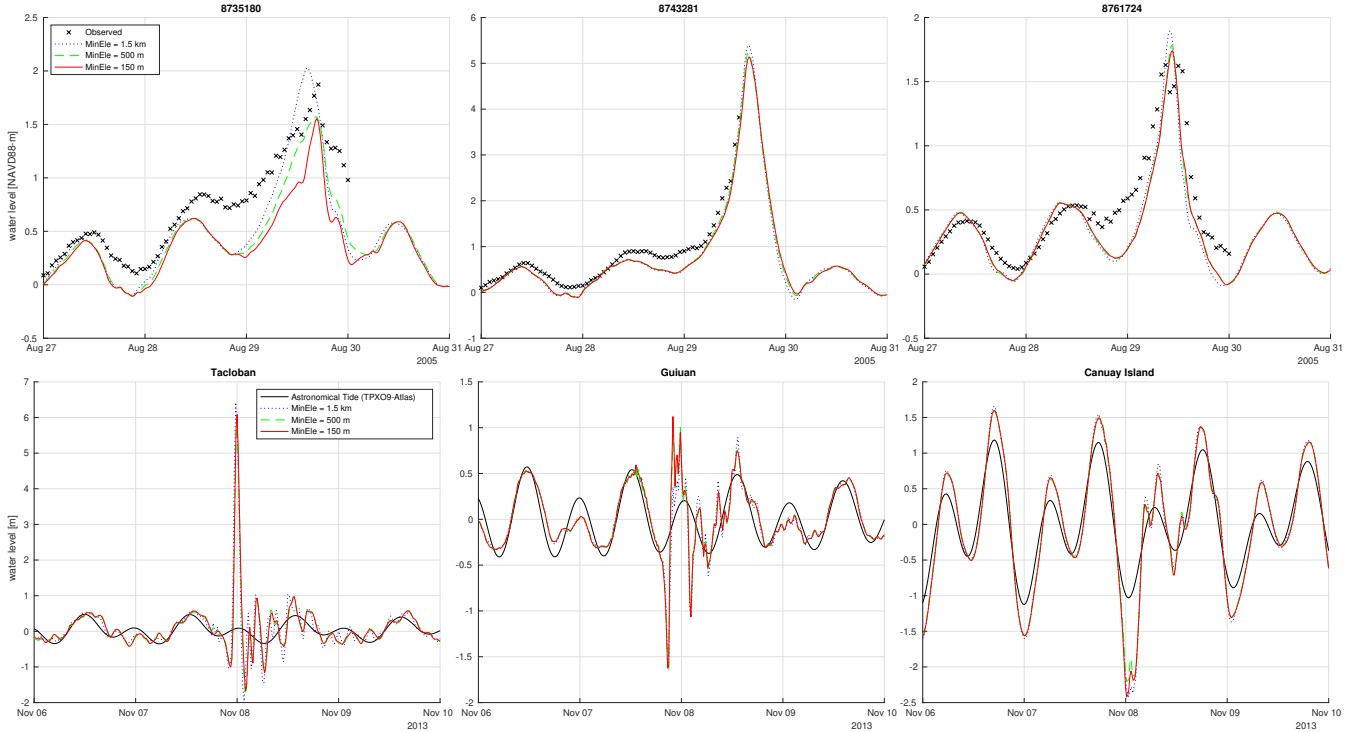

**Figure 11.** Comparisons of simulated storm tide elevation time series on meshes with different local MinEle (1.5 km, 500 m, 150 m) at the point locations shown in Fig 9. Top row: Hurricane Katrina compared to NOAA tide gauge observations (0.23 m has been added to simulated results). Bottom row: Super Typhoon Haiyan compared to the astronomical tide reconstructed from the TPXO9-Atlas.

0.2 m larger on MinEle = 500 m mesh than the MinEle = 150 m mesh (Fig. 12). Interestingly, neither the MinEle = 1.5 km

mesh or the MinEle = 500 m mesh includes Lake Pontchartrain, while the MinEle = 150 mesh does (Fig. 2). This is because the Rigolets strait connecting Lake Pontchartrain to Lake Borgne is approximately 500 m at its narrowest, thus the MinEle = 500 m mesh is at the cutoff point for meshing the strait and providing hydraulic connectivity between the two lakes. Yet, the maximum elevation difference from Lake Borgne across to Mobile Bay between MinEle = 1.5 km and MinEle = 500 m is still much greater than between MinEle = 500 m and MinEle = 150 m (refer time series at gauges 8735180 and 8743281

in Fig. 11 in addition to Fig. 12). Nevertheless, since the MinEle = 150 m is the only mesh to resolve Lake Pontchartrain, the HWMs surrounding the lake (point numbers 7-23) can only be reasonably estimated by simulations on this mesh, resulting in the smallest HWM error statistics (Fig. 10a).

In the case of the Haiyan, the predominant maximum storm tide elevation difference (∼0.4-0.6 m) is located at the back of Leyte Gulf near Tacloban between the MinEle = 1.5 km and MinEle = 500 m meshes (Figs. 11 and 13). The decrease in

elevations in the MinEle = 1.5 km mesh might be explained by the omission of the San Juanico Strait (∼800 m wide at its narrowest, Figure 3). In contrast, the elevations near Tacloban and in the strait increase by up to 0.6 m as the MinEle = 500 m mesh is refined to MinEle = 150 m. There is also a reduction in the elevations (by up to 0.2 m) just offshore of the Leyte Gulf

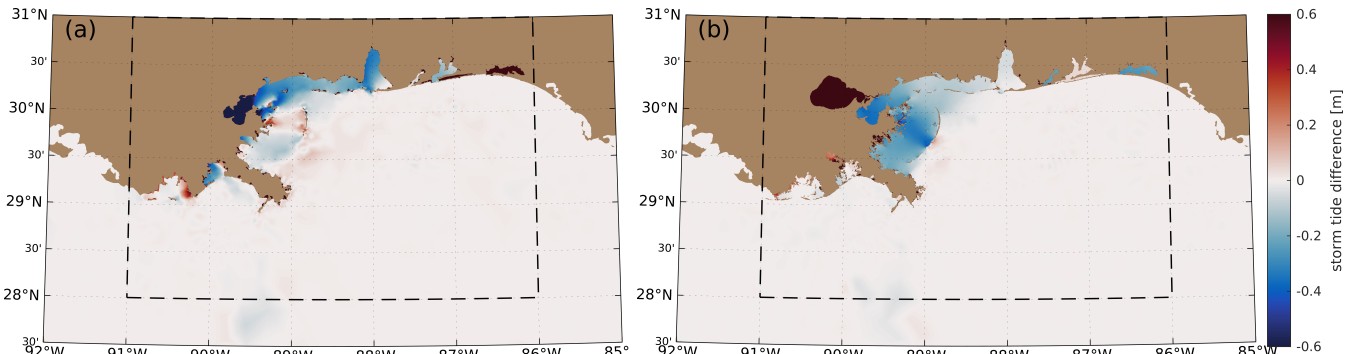

**Figure 12.** Mesh resolution induced differences in the simulated maximum storm tide elevations due to Hurricane Katrina in the Louisiana/Mississippi landfall region. (a) Elevations on MinEle = 500 m local refinement mesh minus elevations on the TLS-B global mesh (MinEle = 1.5 km); (b) Elevations on MinEle = 150 m local refinement mesh minus elevations on the MinEle = 500 m local refinement mesh. In areas where the coarser resolution mesh does not exist the value shown is just the maximum storm tide elevation on the higher resolution mesh.

along the shelf break for the coarser meshes, which is better represented in the higher resolution renditions. Overall, the choice of local MinEle has a small effect on the representation of HWMs for Haiyan (Fig. 10b). The only major noticeable impact is for the HWMs near Tacloban in the San Juanico Strait (point numbers 25-63) on the MinEle = 1.5 km mesh. Since this mesh does not resolve the strait, the simulated estimate of the HWMs here are all taken from the same or nearby mesh vertices at the back of the gulf.

Lastly, we mention two additional general observations. First, for both storms the far-field effects of local mesh refinement were found to be negligible. Second, storm tide elevation time series show that the not only does the peak elevation tend to decrease as mesh refinement is made, but the timing of the peak also tends to occur later (Fig. 11) – most clearly illustrated for gauge 8735180.

### 3.3 Computational Performance

For all astronomical tide simulations performed in the global mesh design experiments (Sect. 3.1) the time step, $\Delta t$, was set to 120 s, which is equivalent to a Courant (CFL) number of 5-22 on the global mesh designs used in this study (see Sect. S1.5 for model stability details). The resulting computational wallclock times for the astronomical tide simulations ranged from 5 s to 30 s per simulation day depending on the total vertex count of the mesh (Fig. 14). All simulations were computed on 480 Haswell processors with a Mellanox FDR Infiniband network connection.

For the storm tide simulations performed in the local mesh refinement experiments (Sect. 3.2) the time step had to be reduced for finer meshes due to model instability related to nonlinear terms that are treated explicitly which become more significant as element sizes nearshore (in shallow depths) are reduced. For Katrina, the time step for the MinEle = 150 m mesh had to be reduced to 40 s. For Haiyan, the time step had to be reduced for both the MinEle = 500 m ($\Delta t$ = 90 s) and the MinEle

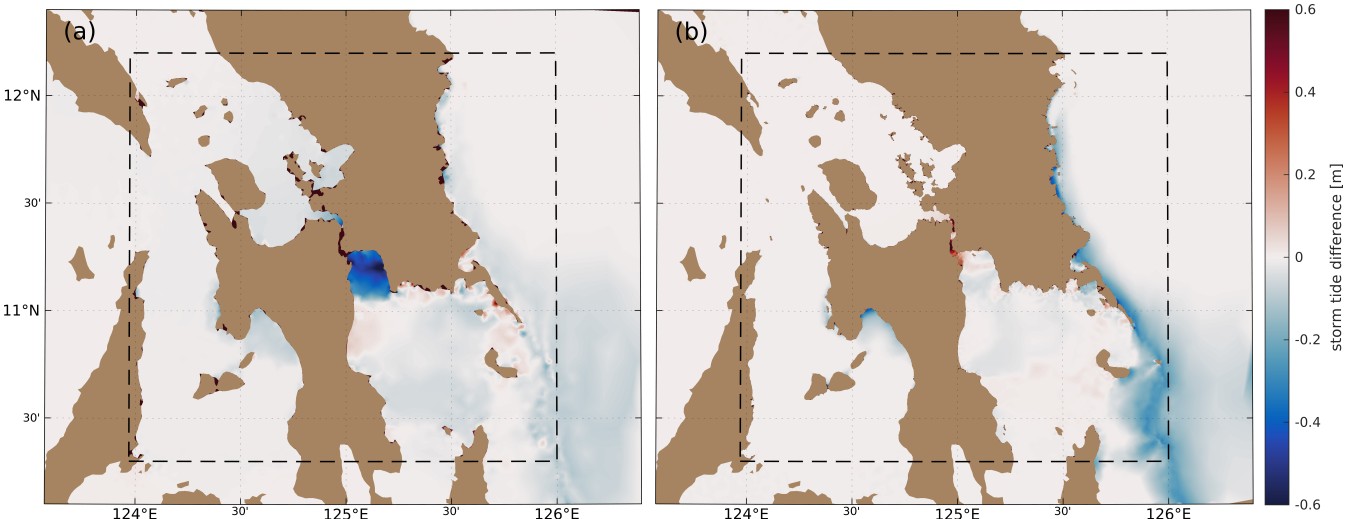

**Figure 13.** Mesh resolution induced differences in the simulated maximum storm tide elevations due to Super Typhoon Haiyan in the Leyte and Samar Island, Philippines landfall region. (a) Elevations on MinEle = 500 m local refinement mesh minus elevations on the TLS-B global mesh (MinEle = 1.5 km); (b) Elevations on MinEle = 150 m local refinement mesh minus elevations on the MinEle = 500 m local refinement mesh. In areas where the coarser resolution mesh does not exist the value shown is just the maximum storm tide elevation on the higher resolution mesh.

= 150 m ($\Delta t$ = 25 s) mesh. The resulting computational wallclock times were increased for the smaller time steps (Fig. 14). However, the simulations on the MinEle = 150 m meshes were proportionally around two times faster per $\Delta t$ compared to the coarser meshes. This is attributed to the heavy I/O related to reading meteorological data during the simulation that limits computational speed-up as time steps are increased beyond a certain value. In fact, for the same $\Delta t$, wallclock times for the storm tide simulations were 2-3 times longer than the astronomical tide runs which require I/O only at the very beginning and end of the simulation. Furthermore, the Haiyan simulations were slightly slower than Katrina simulations because of the higher resolution of the reanalysis meteorology (c.f., Sect. 2.4).

In addition, the computational performance of our Katrina simulations are compared to previous ADCIRC model ones (Bunya et al., 2010; Dietrich et al., 2011), which employed $\Delta t = 1$ s leading to wallclock times one-to-two orders of magnitude greater than those in this study (Fig. 14). We remark that the Bunya et al. (2010); Dietrich et al. (2011) studies used finer mesh sizes (MinEle = 50 m) and included the floodplain leading to more wetting-drying, which can be a source of numerical instability. Our results nonetheless indicate the potential for substantial computational speed-up on suitably designed meshes.

## 4 Discussion

Results from the global mesh design experiments echo those found previously for a regional mesh of the Western North Atlantic Ocean (Roberts et al., 2019b). Namely, that when employing a relatively large element-to-element gradation limit (G)





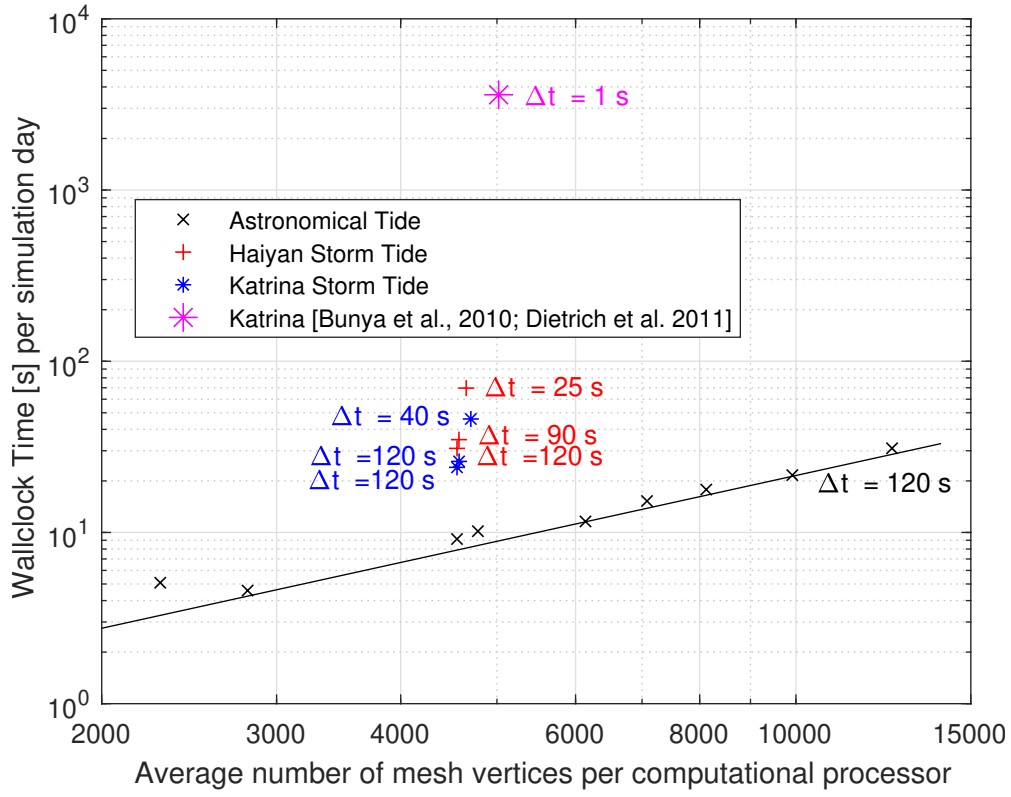

**Figure 14.** Computational wallclock times for the various mesh designs and experiments versus the average number of mesh vertices per computational processor. All simulations were computed on 480 computational processors (i.e., the variation in vertices per processor comes from the variation in the total number of vertices for each mesh design). The computational performance in this study using ADCIRC v55 is contrasted with previous ADCIRC model Katrina storm tide simulations (Bunya et al., 2010; Dietrich et al., 2011) that required very small time steps ($\Delta t = 1$ s).

of 35% used here, the TLS function is critical to ensure that water elevation solutions remain accurate by adequately resolving topographic features of importance. The relatively high value of G ensures that the mesh size rapidly enlarges in areas where no topographic gradient features exist to help avoid excessive mesh vertex counts. An interesting point of difference that we note in this study is the interplay between MinEle and TLS. MinEle not only controls the minimum coastal resolution but it also controls the minimum resolution that can be applied along topographic gradient features in the TLS function. Here, the MinEle ranged between 1.5 km and 6 km which affected the strength of the TLS function in these mesh designs. Nevertheless, the MinEle-C design (MinEle = 6 km) was still superior to the TLS-C one (TLS = 0). There may be additional sensitivity to the TLS function at MinEle less than 1.5 km although it is likely quite small. In Roberts et al. (2019b), MinEle = 1 km was used along the outer shelf and shelf break and this appeared to be sufficient.

The results also show that the use of the barotropic Rossby radius-based low-pass filter (FL) in the TLS function is able to reduce mesh vertices without substantially degrading the solution. However, our results suggest that it is more efficient to





simply reduce the TLS value to 5 compared to using TLS = 20 with FL. Combinations of TLS values between 5-10 with FL (=1/80-1/20) could also be applied depending on the tolerance to the solution accuracy. Using the FL parameter in more highly resolved local domains may also provide benefits to avoid over-resolving topographic slopes in shallow depths. However, it must also be recognized that length scales of importance transition from Rossby radius scaling in the open ocean to those of shoreline features nearshore (LeBlond, 1991). Therefore, a more intelligent low-pass filter may be necessary for such applications.

The astronomical tide solution differences between global mesh designs were shown to be predominantly in the body of the area-weighted ECDF curves while the tails were almost identical. This implies that simulated tides over most of the area of the ocean are affected by mesh resolution. However, in regions where the tidal range and error is large, which inevitably occurs on shallow shelves, all of the mesh designs have similarly large errors. This perhaps explains why a 1/12° tidal model (Schindelegger et al., 2018) could obtain a similar $M_2$ $\overline{\mathrm{RMSE}}_t$ in shallow waters to our more highly resolved reference mesh (14.6 cm compared to 13.9 cm). There is likely an inherent uncertainty in the bathymetry and dissipation which prevents further decrease to the shallow water $\overline{\mathrm{RMSE}}_t$ which can be dominated by large errors (in shallow water the area-weighted median $\mathrm{RMSE}_{t|tot}$ is much smaller than $\overline{\mathrm{RMSE}}_t$). Indeed, the deep ocean $M_2$ $\overline{\mathrm{RMSE}}_t$ was 4.4 cm in Schindelegger et al. (2018), 53% larger than the 2.87 cm on our Ref mesh.

The local mesh refinement technique was demonstrated to be a useful tool to provide high refinement with a trivial addition to the total vertex count. Nevertheless, the numerically stable time step often has to be reduced as coastal mesh resolution becomes finer, which increases computational time. The impact of mesh refinement clearly tends to decrease open ocean storm tide elevations in open ocean areas, but overall the impacts to the HWM errors were relatively small, especially for Super Typhoon Haiyan. The presence of more inland lakes and semi-enclosed bays separated by narrow inlets in the Louisiana region meant that simulated Hurricane Katrina HWM estimates in these features were poor unless the MinEle was sufficient to resolve these narrow passages and provide hydraulic connectivity. A recent study using the GTSM global model with ∼5 km coastal refinement (MinEle = 5 km) was used to characterize the spatiotemporal variability of storm-driven surges along western North Atlantic coasts including surge due to Hurricane Katrina (Muis et al., 2019). Our results would suggest that coarser resolution meshes are be able to reproduce the broader features of surge along open coastal areas quite well. However, such a model cannot be used to predict storm tide levels within the semi-enclosed bays and lakes in the Louisiana region. Furthermore, MinEle = 5 km is above the limit where global astronomical tide solutions appear to diverge noticeably from MinEle = 1.5 km meshes.

Last, it is widely recognized that sensitivities to local high resolution bathymetry datasets, and spatially varying bottom friction and surface ice friction are important (Lefevre et al., 2000; Le Bars et al., 2010; Zaron, 2017; Pringle et al., 2018a; Zaron, 2019), likely more so than the mesh resolution effects that we concentrate on here. We aim to develop a unified framework for calibrating bottom friction globally with improved local high resolution bathymetric datasets in future work. Moreover, we did not simulate inundation in this study, but it will be a crucial future step so that we can forecast and assess coastal flood risks for all of Earth's coasts.



## 5 Conclusions

Important upgrades to the FEM SWE solver, ADCIRC, have been presented to improve accuracy and efficiency for global storm tide modeling across multi-resolution unstructured meshes. We systemically tested the new model's (ADCIRC v55) sensitivity
to mesh design in the simulation of global astronomical tides and storm tides. These mesh design results are expected to be broadly applicable to other SWE solvers that correctly handle solutions on the sphere.

Based on the results for global mesh design we recommend to aim for a minimum element size less than 3 km and to use the TLS function to resolve topographic gradient features with a TLS value of 5-10. Paired with the OceanMesh2D software, the ability to seamlessly apply local refinement allows the user to provide fine coastal resolution in the region of interest (e.g.,
the storm landfall region) without large increases to the total mesh vertex count (increase of 0.5-3% in this study). We found that in general, peak storm tide elevations along the open coast are decreased and the timing of the peak occurs later with local coastal mesh refinement. When validated against observed high water marks measured near the coast, coastal mesh refinement only has a significant positive impact on errors in narrow straits and inlets, and in bays and lakes separated from the ocean by these passages.

The new ADCIRC v55 code capable of accurate global storm tide modeling with fine coastal resolution is computationally efficient. For global meshes with minimum resolution as fine as 1.5 km, the computational wallclock time ranged from 5 s to 30 s per simulation day on 480 computational processors for astronomical tide simulations. Some improvements that we made to the numerical stability of the algorithm facilitated the application of relatively large 120 s time steps to achieve this efficiency. However, we found that the locally refined meshes often required smaller time steps (25 s-90 s). Nevertheless, these
are still much larger than time steps used in previous studies with older versions of ADCIRC, resulting in one-to-two orders of magnitude shorter computational times.

*Code and data availability.*  The official release version of ADCIRC is available from the project website: http://adcirc.org/ under the terms stipulated there and is free for research or educational purposes. The exact version of the model used to produce the results used in this paper is archived on Zenodo (Pringle, 2020) along with the simulated tidal harmonic constituents on the various mesh designs and the simulated
storm tide results with the corresponding model setup (Pringle, 2020).

The current version of OceanMesh2D is available from the project website: https://github.com/CHLNDDEV/OceanMesh2D under the GPL-3.0 license. The exact version of the model (V3.0.0) used to generate the meshes in this study is archived on Zenodo (Pringle and Roberts, 2020).

An application of the presented ADCIRC v55 model (on the TLS-B mesh design) providing 5-day forecasts of global storm surge is
395 currently running in real-time and maximum surge elevations are available to view at https://wpringle.github.io/GLOCOFFS/. The forecast is automatically updated every 6-hours based on GFS weather forecast schedules. Simulation wallclock time is approximately 10 min on 72 computational processors. Images are automatically archived by GitHub.



## Appendix A: Error Metrics

The RMSE$_t$ for a single constituent at a point is defined as (Wang et al., 2012):

$$\text{RMSE}_t = \left(0.5\left[A_o^2 + A_m^2 - 2A_o A_m \cos(\theta_o - \theta_m)\right]\right)^{1/2} \tag{A1}$$

where $A$ is the tidal amplitude and $\theta$ is the tidal phase lag, and the subscripts 'o' and 'm' refer to the observed (in this case TPXO9-Atlas) and modeled values respectively. To get the RMSE$_t$ of the total 5-constituent signal, RMSE$_{t|tot}$, we use the approximation (Wang et al., 2012):

$$\text{RMSE}_{t|tot} = \sum_{k=1}^{5}\left(0.5\left[A_{o_k}^2 + A_{m_k}^2 - 2A_{o_k} A_{m_k} \cos(\theta_{o_k} - \theta_{m_k})\right]\right)^{1/2} \tag{A2}$$

where $k$ indicates the arbitrary constituent number from the five leading tidal constituents ($M_2$, $S_2$, $N_2$, $K_1$, $O_1$).

The area-weighted ECDF curves of RMSE$_{t|tot}$ are computed through MATLAB's "ecdf" function with the 'frequency' input set equal to the elemental area [m$^2$]. With this construction, the area-weighted median value of RMSE$_{t|tot}$ is defined as the point of intersection of the ECDF curve with the 50% probability. Comparisons between ECDF curves can be made by taking the largest vertical distance between them to obtain $K$. We compute $K$ one-sided in that it will be positive if the base (Ref) design is statistically superior and negative otherwise.

To obtain a single number to compare the solutions globally or in certain depths or regions, it is common to use the area-weighted root-mean square error of the RMSE$_t$ (Arbic et al., 2004),

$$\overline{\text{RMSE}_t} = \sqrt{\frac{\iint \text{RMSE}^2\, dA}{\iint dA}} \tag{A3}$$

where $\iint dA$ indicates an area integral. $\overline{\text{RMSE}_t}$ can be computed for a single constituent or for the 5-constituent total signal ($= \overline{\text{RMSE}_{t|tot}}$).

*Author contributions.* WP prepared the manuscript, designed and implemented the coding upgrades into ADCIRC v55, designed and performed the experiments, and conducted the stability analysis. DW mathematically formulated and initially implemented most of the ADCIRC coding upgrades. KR and WP equally contributed to the coding and development of the OceanMesh2D software integral to this study, and KR provided critical feedback to improve the manuscript design. JW provided the research and computing resources and funding necessary to conduct this study.

*Competing interests.* No competing interests.

*Acknowledgements.* This work was supported by the U.S. Alaska/Integrated Ocean Observing System (AOOS/IOOS) subaward for the U.S. National Oceanic and Atmospheric Administration (NOAA) grant #NA16NOS0120027; the IOOS Ocean Technology Transition Project





NOAA grant #NA18NOS0120164; and the NOAA Office of Weather and Air Quality, Joint Technology Transfer Initiative NOAA grant
#NA18OAR4590377.



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
