# Peer review of "Global Storm Tide Modeling with ADCIRC v55: Unstructured Mesh Design and Performance"

_Geoscientific Model Development, 2020_

## Referee Comment (RC1) · Anonymous Referee #1 · 23 Oct 2020

This paper by Pringle et al. presents recent developments of the circulation model ADCIRC that allow simulating efficiently tides and storm surges at global scale. The paper is well-written and organized, the figures are clear and the topic addressed fits well the scope of the journal. However, while storm surge predictions are rather good for a global model, tidal predictions are locally weak compared to other well-established global tidal models. Thus, in the Bay of Biscay, the RMSE on M2 reaches 0.12-0.15 m, that is more than 10% once normalized by the amplitude of this constituent. Over the Patagonian Shelf, RMSE on M2 reaches 0.25 m, which again represents errors over 10 %. In these regions, other global models have errors of a few % in these areas, see for instance a paper describing the hydrodynamic version of FES2014 (i.e. without

assimilation) under discussion in Ocean Science (Lyard et al., 2020). For this reason, I think that the paper cannot be considered further for publication until the authors explain why the model is locally not reproducing tides correctly or better, improve their results. Indeed, only discussing the improvements compared to the previous version of global ADCIRC is not sufficient as tidal predictions from this version of the model were really bad (i.e. errors on M2 locally > 20%).

I would also have the following along-the-text comments:

-L35: I would indicate somewhere that all these studies neglected the contribution of short waves, although this process can drive a "regional setup" (i.e. a storm surge extending outside surf zone) reaching 0.5 m (e.g. Fortunato et al., 2017).

-L73: as the model is used to compute storm surges, you should explain how Cd is computed/which bulk formula is used.

-L100: please explain how much larger

-L106: "obtain" rather than facilitate?

-L157: Gulf of Mexico rather than Western North Atlantic?

-Table 3: please compare with Figure 12 in Lyard et al. (2020), where FES2014 yields errors on M2 < 0.5 cm in deep water and <4 cm on the shelf, that is about one order of magnitude smaller than here.

-L249: as shown by several studies (e.g. Townend and Pethick, 2002) and synthetized in Idier et al., (2019), representing flooding in storm surge models results in lower water levels seaward compared to simulations where the flooding is not represented. Therefore, I expect that water levels in the present simulations are biased high due to this process, possibly by 0.5 to 1.0 m considering previous studies on the topic.

-L256: please refer to Bricker and Roeber (2015) who showed that Hayan also drove very large infragravity waves, which could explain the large scatter on HWMs observed.

-Figure 11: for Katrina, the model displays a 0.5 m negative bias before the surge peak, could the authors comment on the possible causes? Could it be related to the 2DH approach which only allows for a crude representation of Ekman transport?

-L357: please correct "are be able"

-L376: I m not sure that this conclusion is very robust based on a model that does not represent flooding (see my previous comment).

Cited references: Fortunato, A.B., Freire, P., Bertin, X., Rodrigues, M., Ferreira, J. and Liberato, M.L., 2017. A numerical study of the February 15, 1941 storm in the Tagus estuary. Continental Shelf Research 144, 50-64.

Idier, D., Bertin, X., Thompson, P., and Pickering, M., 2019. Interactions between mean sea level, tide, surge, waves and flooding: mechanisms and contributions to sea level variations at the coast. Survey of Geophysics 40, 1603–1630.

Lyard, F. H., Allain, D. J., Cancet, M., Carrère, L., and Picot, N.: FES2014 global ocean tides atlas: design and performances, Ocean Sci. Discuss., https://doi.org/10.5194/os-2020-96, in review, 2020.

Townend, I.H., Pethick, J., 2002. Estuarine flooding and managed retreat. Philos. Trans.R. Soc. Lond. Ser. A 360, 1477–1495.

Roeber, V., Bricker, J.D., 2015. Destructive tsunami-like wave generated by surf beat over a coral reef during Typhoon Haiyan. Nature Communication 6, 7854.

---

## Referee Comment (RC2) · Anonymous Referee #2 · 6 Nov 2020

<General comment> This paper presents the performance of a new version of AD-CIRC on simulating global tides and storm surges, highlighting a mesh design with key parameters identified through experiments, capability of local refinements for extreme events, and improved efficiency brought by updated numerical treatment.

The paper is well organized, and the topic is in line with the scope of GMD. The clear improvement over earlier versions of ADCIRC is surely of interest to existing and potential ADCIRC users. The conclusions and recommendations drawn from the experiments on mesh design and local mesh refinement are valuable for unstructured-grid modelers in general.

There are a few items to be clarified and issues to be addressed (listed below), and my recommendation is "minor revisions".

<Specific comments>

1) Stability constraint

The paragraph starting from Line 303 mentions that smaller time steps are required for locally refined meshes used in the Katrina and Haiyan simulations. How did you decide on an appropriate dt for each simulation? For ADCIRC users, what is an effective way to find the optimal dt for a mesh with local refinements?

Clearly defining the stability condition is generally difficult for complex models, but the users may need a bit more guidance and reference in choosing the time step. If you have additional benchmark tests or applications (done by ADCIRC v55) besides the three configurations mentioned on Line 305-307, please list their grid resolutions and time steps in a table (maybe in the supplemental materials).

Also, consider mentioning the typical grid resolution for global simulations on Line 101. Mention the typical resolution of the refined meshes on Line 384.

2) Solution variability with time step

When model simulations are stable, is there any solution variability with time step? For example, if two simulations are conducted on a same locally refined mesh, one with dt=90 s and another with dt=25 s (values chosen from the suggested range on Line 384), would there be any noticeable difference in the model results (e.g., the timing and elevation of the simulated storm peak)? If not, please add one or two sentences where appropriate to note this.

3) Solution variability with mesh resolution

The effect of mesh resolution on peak elevation and timing is mentioned multiple times in the paper ("Abstract", Section 3.2.2, and "Conclusion"). Do you have any hypothesis

on the mechanism behind this? Could it be that the wave speeds are slightly different due to the difference in model bathymetry (because the resolutions of the mesh are different); or the numerical scheme behaves differently under different Courant numbers?

4) Improved accuracy compared to the prior version

The "Discussion" section focuses on mesh configuration but does not explain the clear improvement between the two model versions on a same ref mesh (Fig. 6ab). Among the numerical improvements from v54 to v55, how does each of them contribute to the improved accuracy (mentioned in Section 3.1.1)? Which one is the main factor? Please add a few sentences or a paragraph to discuss this.

5) Local model error

I agree with Anonymous Referee #1 on that the large local errors (especially those nearshore) need to be discussed and explained, so that the readers/users can have a good understanding of the limitation of this model.

<Technical comments>

Line 29: "FMV" should be "FVM".

Line 252: "match".

Line 357: "are able to".

---

## Author Comment (AC1) · 11 Nov 2020

Dear Reviewer,

Thank you for taking the time to read through our manuscript. We agree on many points that you raised which we will address in a later reply. But first we would like to comment here on the most critical aspect of your review.

The reviewer suggested that since there are a few local shelf regions in the global ocean that have over 10% M2 tidal error that the manuscript should not proceed unless "the authors explain why the model is locally not reproducing tides correctly or

better, improve their results". While we will include additional discussion in a revised manuscript on model properties that affect tide solutions (see next paragraph), we want to first highlight that the major point of our paper is not to present a model with the lowest tidal errors possible. Instead, it is to; 1) highlight improvements to the treatment of the governing equations and implicit time-integration in the new version of ADCIRC (v55), and 2) explore the effects of unstructured mesh design on storm tide solutions. We will state these aims more clearly in a revised version of the manuscript. Through these aims we wish to provide information for other researchers on how to design their own unstructured meshes (for instance, we quantified the critical impact of the topographic-length-scale (TLS) function on mesh resolution distribution) while balancing their tolerance for numerical accuracy vs. their capacity to simulate on meshes of a certain size. Further, by comparing our results to the previous version of ADCIRC we clearly show the impact of incorrectly treating the curvature terms in the governing equations, which is vital for global storm tide modeling.

Aside from further refining model resolution along the coastlines which likely would have a marginal effect on outer shelf and deep ocean solutions, the modeled tidal solutions are dictated by the three major aspects: bathymetry, internal tide wave drag, and bottom friction/bed stress. Both the suggested reference by the reviewer (Lyard et al., 2020), and our own previous study (Pringle et al., 2018) tackles these three aspects in detail. Ultimately, since we decided to focus this study on the effects of mesh resolution and the new ADCIRC version, we deliberately sought to avoid ad-hoc tuning of the internal tide wave drag and bottom friction, and only used the global GEBCO2019 bathymetry database (modified for depths under Antarctic ice shelves). Thus, we only used the typical bottom friction coefficient, $C_f = 0.0025$, everywhere except in the Indian Ocean and Western Pacific Ocean region where we had previously detailed our methodology (Pringle et al., 2018). Similarly, we only calibrated a global tuning coefficient for the internal tide wave drag term. The above decisions are clearly described in the supplementary material, and to prevent ambiguities concerning the main focus of this study we will add a brief summary of this in the main document of a

revised manuscript.

In contrast to our study, Lyard et al., (2020) tunes the model by perturbing regionally disparate values of the internal tide wave drag coefficient and the bottom friction coefficient to obtain tide solutions with smaller errors. Moreover, Lyard et al., (2020) incorporated local bathymetric datasets and perturbed ocean depths in certain polygonal regions to further improve the tidal solutions. As stated in the last paragraph of our discussion, in future work we aim to address, in depth, the bathymetry, internal tide wave drag, and bottom friction/bed stress aspects on global storm tide solutions (in a systematic fashion so that other researchers can reproduce our solutions and efficiently design their own models for their own purpose). In fact, we are already making progress on this front; Figure 1 highlights an example of improved M2 tide errors we obtained by using a couple of local bathymetry datasets and tuning the internal tide wave drag coefficient differently to the current study.

Lastly, we would like to add some context to the reviewer's comment that "tidal predictions are locally weak compared to other well-established global tidal models". To be fair, the well-established tidal model that the reviewer mentions is only one (FES2014; Lyard et al., 2020), in which the manuscript has been under discussion in Ocean Science from October 2020, a few months after we submitted this manuscript. While we acknowledge that the FES model has been under development for a very long time (and our own understanding has greatly benefited from the progress made by that research team), Lyard et al. (2020) is the first study that we have seen clearly showing such small tide errors for the non-assimilated version. Furthermore, while the reviewer points out that the RMSE of the M2 for our model is more than 10% error in the Bay of Biscay and Patagonian Shelf (larger errors than for FES2014 in Lyard et al., 2020); for instance, the M2 RMSE of our model in the Gulf of Guinea (< 2.5 cm) is smaller than that of the non-assimilated FES2014. As stated above, these tidal solutions depend heavily on how one tunes the model. Specifically, our experience tells us that how you tune the internal tide wave drag coefficient in the Atlantic Ocean heavily dictates

the solution in these regions, such that tuning the model to be accurate in the Gulf of Guinea leads to increased errors in the Bay of Biscay, and vice versa. These are the sorts of issues that we are actively researching but which is out of scope for the current study. Nevertheless, the non-assimilated FES2014 model presented in Lyard et al. (2020) does indeed seem to have smaller tide errors overall and we will include it as part of our comparison in a revised manuscript.

Sincerely,

William Pringle and co-authors.

References:

Lyard, F. H., Allain, D. J., Cancet, M., Carrère, L., and Picot, N.: FES2014 global ocean tides atlas: design and performances, Ocean Sci. Discuss., https://doi.org/10.5194/os2020-96, in review, 2020.

Pringle, W. J., Wirasaet, D., Suhardjo, A., Meixner, J., Westerink, J. J., Kennedy, A. B., & Nong, S. (2018). Finite-Element Barotropic Model for the Indian and Western Pacific Oceans: Tidal Model-Data Comparisons and Sensitivities. Ocean Modelling, 129, 13–38. https://doi.org/10.1016/j.ocemod.2018.07.003.

Case C8: M2 RMSE

**Fig. 1.** M2 tide RMSE computed against the TPXO9-Atlas for a modified version of the model presented in the current study. The model shown here has smaller errors than shown in Fig. 6b of the manuscript.

---

## Author Comment (AC2) · 8 Dec 2020

Dear Reviewer,

Thank you again for taking the time to read through and comment on our manuscript. We responded to your greatest criticism in a previous reply. In this comment, we give a point-by-point response to all your comments and detail our proposed changes to the manuscript.

R: Reviewer's comment

A: Author's response

C: Proposed changes to the manuscript; text changes in blue

[All references that we cite herein can be found in the reference list of the modified manuscript.]

\_\_\_\_\_

R: This paper by Pringle et al. presents recent developments of the circulation model ADCIRC that allow simulating efficiently tides and storm surges at global scale. The paper is well-written and organized, the figures are clear and the topic addressed fits well the scope of the journal. However, while storm surge predictions are rather good for a global model, tidal predictions are locally weak compared to other well-established global tidal models. Thus, in the Bay of Biscay, the RMSE on M2 reaches 0.12-0.15 m, that is more than 10% once normalized by the amplitude of this constituent. Over the Patagonian Shelf, RMSE on M2 reaches 0.25 m, which again represents errors over 10 %. In these regions, other global models have errors of a few % in these areas, see for instance a paper describing the hydrodynamic version of FES2014 (i.e. without assimilation) under discussion in Ocean Science (Lyard et al., 2020). For this reason, I think that the paper cannot be considered further for publication until the authors explain why the model is locally not reproducing tides correctly or better, improve their results. Indeed, only discussing the improvements compared to the previous version of global ADCIRC is not sufficient as tidal predictions from this version of the model were really bad (i.e. errors on M2 locally > 20%).

A: Thank you for your positive comments regarding the general organization and presentation of this manuscript. As per our previous response, we highlight that the major point of our paper is not to present a model with the lowest tidal errors possible. Instead, it is to; 1) highlight improvements to the treatment of the governing equations and implicit time-integration in the new version of ADCIRC (v55), and 2) explore the effects of unstructured mesh design on storm tide solutions. In the previous response we also highlighted that modeled tidal solutions are dictated by the three major mechanisms: bathymetry, internal tide wave drag, and bottom friction/bed stress. Analysis of these solution controlling mechanisms have been detailed in previous studies (Lyard et al., 2020; Pringle et al., 2018), and in this study we specifically avoided the excessive tuning of the model through these three controls. The previous response also provides an example figure of the M2 tidal solution errors (errors are generally smaller than presented in this manuscript) of a more tuned version of the model used in this study. In the next paragraph we detail our proposed changes to the manuscript to "explain why the model is locally not reproducing tides correctly" and state our aims and decisions more clearly.

C:

1) We explicitly state the aims of the study at the end of "Section 1: Introduction", Lines 56-58:

Section 3.3 summarizes the timing results with ADCIRC v55, highlighting its computational efficiency using a semi-implicit time-integration scheme. In summary, this study aims to: 1) highlight improvements to the treatment of the governing equations and implicit time-integration in the new version of ADCIRC (v55), and 2) explore the effects of unstructured mesh design on storm tide solutions.

2) In "Section 2.4: Datasets and Model Setup". At the end of the first paragraph which specified the bathymetric data used as well as other data use in the model setup, we propose to add the following sentences that outline how we specified the bottom friction and the internal wave drag coefficients and the reasons for this (Lines 193-201): We note here that the accuracy of global tidal solutions strongly depends on the quality of the bathymetric data, the internal wave drag tensor, and the bottom stress term which can all be tuned to minimize tidal errors (Pringle et al., 2018a, Lyard et al., 2020). Since this study is focused on the effects of mesh design and the improvements to the governing

equations in the new version of the ADCIRC model, we deliberately avoided excessive tuning of the model with the aim to minimize tidal solution errors. Instead we chose to use a global constant value of Cit which gives the same available potential tidal energy as compared to the TPXO9-Atlas, and employ a global constant Cf of 0.0025 except in the Indian Ocean and Western Pacific Ocean where it is spatially varying per the specifications by a previous study of ours (Pringle et al., 2018a) (see Sect. S2 for additional details of model specifications).

**3) We included additional details on the model properties that affect tide solutions into "Section 4: Discussion":**

Lines 378-380:

Indeed, a recent study conducts a 432-member ensemble of perturbations to bathymetric depths, and bottom friction and internal wave drag coefficients to obtain smaller tidal errors than this study, particularly in shallow water (Lyard et al, 2020). Lines 402-407:

Last, it is widely recognized that sensitivities to local high resolution bathymetry datasets, internal tide wave drag, and spatially varying bottom friction and surface ice friction are important (Lefevre et al., 2000; Le Bars et al., 2010; Zaron, 2017; Pringle et al., 2018a; Zaron, 2019; Lyard et al., 2020) likely more so than the mesh resolution effects that we concentrate on here. We aim to develop a unified framework for globally calibrating spatially varying internal tide wave drag and bottom friction coefficients with improved local high resolution bathymetric datasets in future work. Doing so should result in smaller storm tide elevation discrepancies especially in shallow water (e.g., Lyard et al., 2020).

R: -L35: I would indicate somewhere that all these studies neglected the contribution of short waves, although this process can drive a "regional setup" (i.e. a storm surge extending outside surf zone) reaching 0.5 m (e.g. Fortunato et al., 2017).

A: We added the following sentence following the citation to these previous studies on extreme sea levels.

C: Line 35-37: Note that these previous studies neglected the contributions to extreme sea levels by short waves that can drive a significant regional setup (e.g., Fortunato et al., 2017).

**R: -L73: as the model is used to compute storm surges, you should explain how Cd is computed/which bulk formula is used.**

A: We define Cd on Line 84 (old L83) so we added the drag law formulation information to that line.

C: Line 87 (old L83): ... computed using the Garratt (1977) drag law.

**R: -L100: please explain how much larger**

A: This information (Courant number = 5-22 with 120 s time step) is contained within "Section 3.3 Computational Performance" so we modified the sentence to refer the reader to this section for details.

C: Lines 105-107 (old L100): With a semi-implicit time integration scheme, the computational time step permitted is larger than the CFL constraint and as a result facilitates computationally efficient global simulations (see Sect. 3.3 for details).

**R: -L106: "obtain" rather than facilitate?**

C: Line 111 (old L106): changed to obtain

**R: -L157: Gulf of Mexico rather than Western North Atlantic?**

A: The Western North Atlantic here refers to one of the basins where tropical cyclones form and we are including the Gulf of Mexico within that definition of Western North Atlantic. Therefore, we decided not to edit this.

R: -Table 3: please compare with Figure 12 in Lyard et al. (2020), where FES2014 yields errors on M2 < 0.5 cm in deep water and <4 cm on the shelf, that is about one order of magnitude smaller than here.

A: We agree to compare with hydrodynamic FES2014 (Lyard et al., 2020) here but we take numbers from Table 1 in Lyard et al. (2020) which gives the overall RMS of the vector difference which we can compare to our numbers shown in the Table 3 (the errors are

greater than the reviewer states). We also realize that the Ngodock et al. (2016); Schindelegger et al. (2018); Lyard et al. (2020) are computing errors only for latitudes equatorward of  $\pm 66^{\circ}$ , so we included our result for within these latitudes as well (results are not that different). We also add some two sentences to "Section 3.1.1: Validation of the Reference Mesh" commenting on the comparison to the FES2014 results. C:

**1) New Table 3:**

**Table 3.**  $\overline{\text{RMSE}}_{\ell}$  [cm] (c.f., Appendix A) values for simulated tidal results using ADCIRC v55 (upgrade) and ADCIRC v54 in deep (h > 1 km) and shallow (h < 1 km) waters on the Ref mesh. Results from other forward barotropic tidal models (Stammer et al., 2014; Ngodock et al., 2016; Schindelegger et al., 2018; Lyard et al., 2020) are included for comparison where known.

|                              |                       | $M_2 \overline{RMSE}_t [cm]$ |               | $\overline{\text{RMSE}}_{t tot}$ [cm] |               |
|------------------------------|-----------------------|------------------------------|---------------|---------------------------------------|---------------|
| Model                        | Latitudes             | Deep water                   | Shallow water | Deep water                            | Shallow water |
| ADCIRC v54                   | All                   | 6.5                          | 18.5          | 7.92                                  | 22.1          |
| ADCIRC v55                   | All                   | 2.87                         | 13.9          | 3.89                                  | 17.2          |
| ADCIRC v55                   | $\leq \pm 66^{\circ}$ | 2.85                         | 14.7          | 3.81                                  | 18.2          |
| Stammer et al. (2014)*       | All                   | 5.25-7.76                    | 18.6-27.9     | -                                     |               |
| Ngodock et al. (2016)*#      | $\leq \pm 66^{\circ}$ | 2.6-3.2                      | -             | -                                     | =             |
| Schindelegger et al. (2018)* | $\leq \pm 66^{\circ}$ | 4.4                          | 14.6          | -                                     | -             |
| Lyard et al. (2020)**        | $\leq \pm 66^{\circ}$ | 1.53                         | 6.44          | -                                     | -             |

\*: RMSEt is computed against TPXO8-Atlas rather than TPXO9-Atlas.

**: Uses state ensemble Kalman Filter (perturbed data assimilation).**

\*\*: RMSEt results for hydrodynamic FES2014 computed against satellite cross-over points.

2) Modified Lines 225-234 (end of Section 3.1.1): The deep ocean M2 RMSEt = 2.87 cm (Table 3) is smaller than for the majority of previously non-assimilated barotropic tidal models (Stammer et al., 2014; Schindelegger et al., 2018), and within the range of errors computed for solutions obtained by embedding a state ensemble Kalman Filter (perturbed data assimilation) into a forward ocean circulation model (Ngodock et al., 2016). The recent study by Lyard et al. (2020) carefully tunes local bathymetric data and dissipation parameters (Cf and Cit) to obtain smaller errors (M2 RMSEt = 1.53 cm) than presented here. As noted in Sect. 2.4, in this study we deliberately avoided excessive tuning of the model with the aim to minimize tidal solution errors. Nevertheless, the 5-constituent total tidal error, RMSEt|tot, is less than 4 cm in the deep ocean. In shallow regions, the M2 RMSEt is 13.9 cm, which is essentially the same as presented in Schindelegger et al. (2018), but significantly greater than in Lyard et al. (2020). The total tidal error in shallow water, RMSEt|tot is 17.2 cm, but note that the area-weighted median value of shallow water RMSEt|tot (c.f. Appendix A) is just 6.63 cm.

R: -L249: as shown by several studies (e.g. Townend and Pethick, 2002) and synthetized in Idier et al., (2019), representing flooding in storm surge models results in lower water levels seaward compared to simulations where the flooding is not represented. Therefore, I expect that water levels in the present simulations are biased high due to this process, possibly by 0.5 to 1.0 m considering previous studies on the topic. A: We agree with the point that you raised regarding that including inundation in the simulation would result in lower water levels seaward. In other words, the maximum coastal water levels shown in Figure 9 are likely biased high. However, here we are comparing to high water marks (HWM) measured on land using the closest modeled wet point. Runup onto the land can amplify the water levels beyond those recorded seaward, but since our simulated results at the coast are likely biased high there is a degree of cancellation involved. Nevertheless, we noted that our closest wet point results might only follow the lower envelope of HWMs as noted by Mori et al. (2014) since the amplification could be greater than the low bias due to not simulating inundation especially in the presence of steep topography.

C: Added to the end of Lines 267-268 (old L249): ... (although ignoring inundation in our simulations is expected to overestimate the seaward maximum storm tide heights (Idier et al., 2019) that likely cancels out some of the otherwise low bias when compared to HWMs).

R: -L256: please refer to Bricker and Roeber (2015) who showed that Hayan also drove very large infragravity waves, which could explain the large scatter on HWMs observed.
A: Thank you, it is a good idea to point out the large scatter in the HWM measurements and this potentially being related to infragravity wave generation over reefs.
C: Added to end of paragraph on Lines 274-276 (after old L256): The large scatter present in the HWM measurements (SD ≈ 1.3 m for all MinEle) could be related to the generation of infragravity waves over fringing reefs in the region leading to amplified coastal runup (Roeber and Bricker, 2015).

R: -Figure 11: for Katrina, the model displays a 0.5 m negative bias before the surge peak, could the authors comment on the possible causes? Could it be related to the 2DH approach which only allows for a crude representation of Ekman transport?

A: On lines 261-264 of the original manuscript we noted that this negative bias could have been due to the neglect of the regional wave setup since a previous ADCIRC-based study that coupled to short waves better matched the time series before the surge peak (Roberts and Cobell, 2017). However, after subsequent simulations by our group on separate but related research, we do not think that this bias is mostly attributable to the insufficient generation of the surge forerunner (e.g., Kennedy et al., 2011). This fact indeed arises from the crude representation of Ekman transport by the 2DH approach as the reviewer surmises, but the negative effect can be mitigated by setting the bottom friction coefficient to a very small value on the shelf. The previous studies by Bunya et al. (2010) and Roberts and Cobell (2017) used a Manning's formulation for the bottom friction coefficient (where n ~ 0.02 in the ocean) which leads to very small values of the Cf on the continental shelf (~50-200 m deep).

C: Changed the old lines 261-264 to the following (new Lines 283-289): We think that this negative bias is mostly attributable to the insufficient generation of the surge forerunner and partly also to the omission of regional wave setup. The surge forerunner is generated through the Ekman setup process (Kennedy et al., 2011) which is crudely represented by the depth-averaged model used here. Previous depth-averaged ADCIRC-based studies that used a Manning's bottom friction formulation so that Cf becomes very small on the continental shelf appear to be better able to generate the surge forerunner, as well as employing wind wave-coupling that generates wave setup, indeed show better agreement with the time series prior to the peak storm tide (Bunya et al., 2010; Roberts and Cobell, 2017).

**R: -L357: please correct "are be able"**

C: Line 398 (old L357): ... are able to ...

**R: -L376: I'm not sure that this conclusion is very robust based on a model that does not represent flooding (see my previous comment).**

A: We agree with your comment that when including inundation in the simulation the seaward maximum storm tide heights would be decreased. So, we modified parts of paragraph in paragraph 4 of "Section 4: Discussion" in addition to this line in "Section 5: Conclusions" to comment on this potential effect, noting that the coarser models would have a greater coastal flooding potential.

C:

1) Lines 389-393: In practice, higher peak storm tide heights in coarser models translates to greater coastal flooding potential. Including inundation in the model would decrease the storm tide elevations along the coast (Idier et al., 2019) perhaps leading to more similar coastal storm tide elevations between the different mesh resolutions since more flooding may occur in the coarser model. Overall, the impacts of mesh resolution on the HWM errors were relatively small, especially for Super Typhoon Haiyan. However, the ...

2) Line 419 (Old L376): We found that in general, peak storm tide elevations along the open coast are decreased (therefore the coastal flooding potential is decreased) ...

---

## Author Comment (AC3) · 8 Dec 2020

Dear Reviewer,

Thank you for taking the time to read through and comment on our manuscript. Here, we give a point-by-point response to all your comments and detail our proposed changes to the manuscript.

R: Reviewer's comment
A: Author's response
C: Proposed changes to the manuscript; text changes in blue

[All references that we cite herein can be found in the reference list of the modified manuscript.]
* * *
**R: This paper presents the performance of a new version of ADCIRC on simulating global tides and storm surges, highlighting a mesh design with key parameters identified through experiments, capability of local refinements for extreme events, and improved efficiency brought by updated numerical treatment. The paper is well organized, and the topic is in line with the scope of GMD. The clear improvement over earlier versions of ADCIRC is surely of interest to existing and potential ADCIRC users. The conclusions and recommendations drawn from the experiments on mesh design and local mesh refinement are valuable for unstructured-grid modelers in general. There are a few items to be clarified and issues to be addressed (listed below), and my recommendation is "minor revisions"**

A: Thank you for your positive comments. We aim to fully address each of your identified issues as detailed below.

**R: Stability constraint:**
**The paragraph starting from Line 303 mentions that smaller time steps are required for locally refined meshes used in the Katrina and Haiyan simulations. How did you decide on an appropriate dt for each simulation? For ADCIRC users, what is an effective way to find the optimal dt for a mesh with local refinements?**
**Clearly defining the stability condition is generally difficult for complex models, but the users may need a bit more guidance and reference in choosing the time step. If you have additional benchmark tests or applications (done by ADCIRC v55) besides the three configurations mentioned on Line 305-307, please list their grid resolutions and time steps in a table (maybe in the supplemental materials).**
**Also, consider mentioning the typical grid resolution for global simulations on Line 101. Mention the typical resolution of the refined meshes on Line 384.**
A: We agree that it would be very helpful to know what the "optimal" dt would be for a certain mesh. In the supplementary we conducted the stability analysis to determine what is the stability constraint under the linear 1-D conditions (Sect. S1.5). However, this only provides us knowledge of the maximum value of the product of dt and the numerical parameter, tau0 (for linear stability). Thus, the dt we set for each of the tropical cyclone simulations was actually found by trial-and-error while keeping the ratio of dt and tau0 constant (see Sect. S2.1).
The reason that higher resolution meshes may need a smaller time step could be related to the CFL condition based on the fluid velocity (instead of the gravity wave speed). Considering the maximum fluid velocity *a priori* is not obvious but if we back out the results and consider Cr = 1 to be the stability criteria then for the two tropical cyclone tests we have (note that the values of dx listed are the actual minimum element edgelengths of the mesh while MinEle is the nominal minimum resolution. The stable dt for each mesh is listed in the Sect. S2.1):

$\qquad$ Cr = U*dt/dx = > U = Cr*dx/dt:
- MinEle = 1.5 km (both)    : U = 1*300 m / 120 s = 2.5 m/s
- MinEle = 500 m (Haiyan)  : U = 1*98.4 m / 80 s  = 1.3 m/s
- MinEle = 500 m (Katrina) : U = 1*135 m / 120 s  = 1.1 m/s
- MinEle = 150 m (Haiyan)  : U = 1*64.2 m / 30 s  = 2.2 m/s
- MinEle = 150 m (Katrina) : U = 1*91.1 m / 50 s  = 1.8. m/s

The value of U to use in the CFL condition varies by around a factor of 2 between the meshes but this is at least superior to an order of magnitude estimate. Based on the results of these tests, setting U ≈ 2.5 m/s in the CFL condition may be a reasonable guideline to determining a stable dt for ADCIRC simulations using the semi-implicit time

integration. However, this is only a guideline and surely does not guarantee stability. It is also unclear how well this will translate into simulations with more wetting-drying.

C:

1) We added this information on velocity-based CFL criteria listed above into the supplementary Sect. S2.1, lines 199-211:

5. dt is set to approximately the largest value that enables reliably stable simulations based on experience and trial-and-error. Although the linear CFL condition is satisfied unconditionally, nonlinear terms introduce instabilities on finer meshes in shallow depths, and could be related to the CFL condition based on the fluid velocity (instead of the gravity wave speed), i.e., $Cr = U_{max}*dt/dx$. dt = 120 s was used for all simulations on the global mesh without local refinement, while the stable dt was generally smaller for the storm tide simulations on the meshes with local refinement. Hurricane Katrina: dt = 120 s on the MinEle = 500-m mesh, and dt = 50 s on the MinEle = 150-m mesh. Super Typhoon Haiyan: dt = 80 s on the MinEle = 500-m mesh, and dt = 30 s on the MinEle = 150-m mesh. Based on these results and rearranging the CFL condition for the maximum fluid velocity, $U_{max}$ with Cr set to 1 as the stability criteria and using the actual minimum element edgelengths of the mesh (rather than the nominal minimum resolution, MinEle) we obtain $U_{max}$ = 1.1-2.5 m/s. Therefore, setting $U_{max}$ to 2.5 m/s in the fluid velocity-based CFL condition could be used as a guideline for determining a stable dt for ADCIRC simulations using the semi-implicit time integration. However, this is only a guideline and does not guarantee stability. The corresponding ADCIRC `fort.15' control file parameter for dt is DTDP (https://wiki.adcirc.org/wiki/DTDP).

2) We modified Lines 382-383 (old line 349) to add the reference to Sect. S2.1:

Nevertheless, we found that the numerically stable time step decreases as coastal mesh resolution becomes finer (see Sect. S2.1 for details on setting the time step), which increases computational time.

2) Lines 105-107 (old line 101): With a semi-implicit time integration scheme, the computational time step permitted is larger than the CFL constraint and as a result facilitates computationally efficient global simulations on meshes that have nominal minimum resolutions of 150 m-1.5 km (see Sect 3.3 for details).

3) Line 427 (old line 384): (nominal minimum resolution of 500 m and 150 m)

R: Solution variability with time step:

**When model simulations are stable, is there any solution variability with time step? For example, if two simulations are conducted on a same locally refined mesh, one with dt=90 s and another with dt=25 s (values chosen from the suggested range on Line 384), would there be any noticeable difference in the model results (e.g., the timing and elevation of the simulated storm peak)? If not, please add one or two sentences where appropriate to note this.**

A: From a limited additional test set we do not see noticeable differences between solutions using different time steps since even the larger 120 s time step is still much smaller than the period of the shallow water waves. We also verified that ADCIRC temporal discretization errors are much smaller than spatial discretization errors in Roberts et al. (2019b) [Section 3.5.2 of that paper]. Variations due to the time step could nevertheless become more apparent when significant wetting-drying occurs since the methodology used in ADCIRC assumes that only one dry element adjacent to a wet element can become wet (and vice-versa for drying) per computational time step.

C: Added this note in paragraph 4 of "Section 4: Discussion" Lines 383-386: Note that additional tests (not shown) were conducted, and these demonstrated that the computational time step used for the same mesh had a negligible effect on storm tide elevation solutions. However, this may not transfer as well for simulations where there is significant wetting-drying due to the one element per time step wetting-drying logic used.

R: Solution variability with mesh resolution:

**The effect of mesh resolution on peak elevation and timing is mentioned multiple times in the paper ("Abstract", Section 3.2.2, and "Conclusion"). Do you have any hypothesis on the mechanism behind this? Could it be that the wave speeds are slightly different due to the difference in model bathymetry (because the resolutions of the mesh are different); or the numerical scheme behaves differently under different Courant numbers?**

A: We think that this is primarily related to the bathymetry and the geometric representation of shoreline boundary (physical approximation errors) which results in the

slightly different wave speeds and wave transformation as the reviewer mentions. Our previous work presented in Roberts et al. (2019b) details these effects. For instance, we showed that the error of the polygonal area of the mesh increases geometrically as the minimum shoreline resolution is coarsened [Figure 4 of Roberts et al. (2019b)]. Similarly, we also showed that the volumetric error of the mesh increases geometrically with mesh coarsening [Figure 7 of Roberts et al. (2019b)]. Section 3.5.2 of Roberts et al. (2019b) discusses the relative make-up of numerical discretization errors versus the aforementioned physical approximation errors for astronomical tide solutions. The findings concluded that ADCIRC numerical discretization errors are non-trivial but generally less significant than the physical approximation errors associated with mesh refinement/coarsening along shorelines and topographic gradients.

C: We added a small note on this to "Section 4: Discussion" on Lines 386-389: The impact of mesh refinement clearly tends to decrease open ocean storm tide elevations in open ocean areas and the timing of the peak occurs later. This could be attributed to larger physical approximation errors of the shoreline geometry and bathymetry with mesh coarsening (c.f. Roberts et al., 2019b) leading to slightly different wave speeds and wave transformation.

**R: Improved accuracy compared to the prior version:**
**The "Discussion" section focuses on mesh configuration but does not explain the clear improvement between the two model versions on a same ref mesh (Fig. 6ab). Among the numerical improvements from v54 to v55, how does each of them contribute to the improved accuracy (mentioned in Section 3.1.1)? Which one is the main factor? Please add a few sentences or a paragraph to discuss this.**

A: The reason for the improvement is due to the changes to the governing equations which is discussed in detail in supplementary Sect. S1.2. To summarize, the form of the equations (particularly the continuity equation) solved in the old version of ADCIRC did not correctly consider the curvature on the spherical Earth, so it was technically only valid for "small" domains.

C: We added a paragraph to the beginning of "Section 4: Discussion" to make this clear (Lines 345-351): The new version of ADCIRC (v55) demonstrated improved tidal solutions compared to the previous versions of ADCIRC (denoted as ADCIRC v54). This is because ADCIRC v54 does not solve the correct form of the governing equations in Spherical coordinates and is thus technically valid only for sufficiently small regional domains (see Sect. S1.2 for more details on this comparison). For instance, this old form of the governing equations appears to be sufficient for the western North Atlantic Ocean regional domain, which has been thoroughly validated using ADCIRC since Westerink et al. (1994). The changes made in ADCIRC v55 make it suitable for simulating larger domains, in particular the global domains that we investigated in this study.

**R: Local model error:**
**I agree with Anonymous Referee #1 on that the large local errors (especially those nearshore) need to be discussed and explained, so that the readers/users can have a good understanding of the limitation of this model.**

A: Please refer to our first reply to Anonymous Referee #1 for a detailed response to this comment. In essence, we do not think that the presence of some larger local errors is a result of a structural error of the model. Instead, such errors are predominantly related to a combination of imperfect bathymetric data and dissipation approximations that can be "tuned" in a way to further reduce the error. We also added some lines throughout the manuscript to address this as detailed in our second reply to Anonymous Referee #1.

**R: Technical corrections:**
**Line 29: "FMV" should be "FVM".**
**Line 252: "match".**
**Line 357: "are able to".**
C: Made these corrections.